# WARPD: World model Assisted Reactive Policy Diffusion

## Abstract

With the increasing availability of open-source robotic data, imitation learning has become a promising approach for both manipulation and locomotion. Diffusion models are now widely used to train large, generalized policies that predict controls or trajectories, leveraging their ability to model multimodal action distributions. However, this generality comes at the cost of larger model sizes and slower inference, an acute limitation for robotic tasks requiring high control frequencies. Moreover, Diffusion Policy (DP), a popular trajectory-generation approach, suffers from a trade-off between performance and action horizon: fewer diffusion queries lead to larger trajectory chunks, which in turn accumulate tracking errors. To overcome these challenges, we introduce WARPD (World model Assisted Reactive Policy Diffusion), a method that generates closed-loop policies (weights for neural policies) directly, instead of open-loop trajectories. By learning behavioral distributions in parameter space rather than trajectory space, WARPD offers two major advantages: (1) extended action horizons with robustness to perturbations, while maintaining high task performance, and (2) significantly reduced inference costs. Empirically, WARPD outperforms DP in long-horizon and perturbed environments, and achieves multitask performance on par with DP while requiring only $\sim 1/45th$ of the inference-time FLOPs per step.

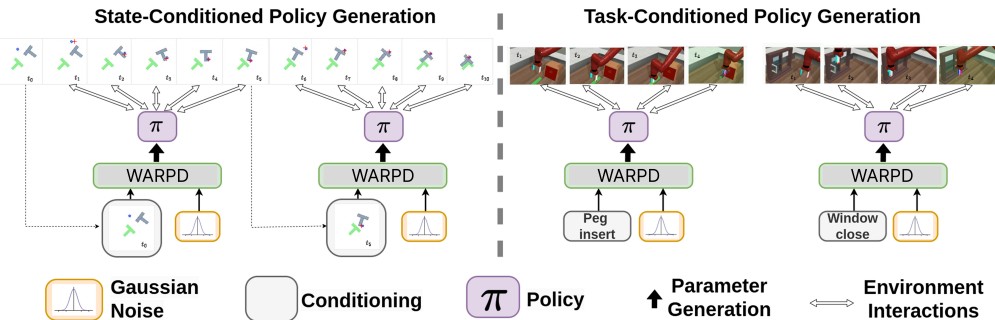

Figure 1: **WARPD** generates policies from heterogeneous trajectory data. With state-conditioned policy generation, the diffusion model can run inference at a lower frequency. With task-conditioned policy generation, the generated policies can be small yet maintain task-specific performance. Demonstrations of this work can be found on the project website: `https://sites.google.com/view/warpd/home`.

## 1 Introduction

The rise of open-source robotic datasets has made imitation learning a promising approach for robotic manipulation and locomotion tasks (Collaboration et al., 2023; Peng et al., 2020). While methods like Behavioral Cloning (Florence et al., 2022) and transformer-based models (e.g., RT-1 (Brohan et al., 2022)) have shown promise, they struggle with multimodal action distributions. For example, in navigation tasks where both "turn left" and "turn right" are valid, these models often predict an averaged action, i.e., "go straight", leading to suboptimal performance.

Diffusion models offer a compelling alternative, providing continuous outputs and learning multimodal action distributions (Tan et al., 2024). Action trajectory diffusion for robotic tasks (Chi et al., 2024) has shown promise but incurs high computational costs, particularly at high control

frequencies. Moreover, such trajectory diffusion models are susceptible to the trade-off between performance and action horizon (or action chunk size, representing the number of environment interactions between consecutive trajectory generations). Fewer diffusion queries lead to larger action chunks, giving greater trajectory tracking errors.

To overcome these limitations, we introduce **World model Assisted Reactive Policy Diffusion (WARPD)**, a novel approach that uses latent diffusion and a world model to **generate closed-loop policies directly in parameter space**, bypassing trajectory generation. WARPD first encodes demonstration trajectories into a latent space, then learns their distribution using a diffusion model, and finally decodes them into policy weights via a hypernetwork (Ha et al., 2016). The generated policy is also optimized with model-based imitation learning using a co-trained world (dynamics) model (Ha & Schmidhuber, 2018), which helps in understanding the environment transitions during training. This approach leverages the success of latent diffusion techniques in vision (Rombach et al., 2022b) and language (Lovelace et al., 2024), and combines them with learned dynamics models, bringing their advantages to robotic control. The world model, and accompanying loss terms, help the agent learn the optimal policy that can be backpropagated through the learned (differentiable) dynamics, and also apply corrective actions to bring the agent states back into the distribution of the input trajectory dataset. For WARPD, the action horizon corresponds to the number of environment interactions between consecutive policy weight generations. To achieve trajectory encoding and policy parameter decoding, we derive a novel objective function described in section 3.3, and show that we can approximate its components with a hypernetwork-based VAE and a World Model, and optimize it using a novel loss function described in section 3.2. This paper provides the following key contributions:

1. **Theoretical Foundations for generating policies**: By integrating concepts from latent diffusion, hypernetworks, and world models, we derive a novel objective function, which when optimized, allows us to generate policy parameters instead of action trajectories.

2. **Longer Action Horizons & Robustness to Perturbations**: By generating closed-loop policies under learned dynamics, WARPD mitigates trajectory tracking errors, enabling policies to operate over extended time horizons with fewer diffusion queries. Additionally, closed-loop policies are reactive to environmental changes, ensuring WARPD-generated policies remain robust under stochastic disturbances.

3. **Lower Inference Costs**: The computational burden of generalization is shifted to the diffusion model, allowing the generated policies to be smaller and more efficient.

We validate these contributions through experiments on the PushT task (Chi et al., 2024), the Lift and Can tasks from Robomimic (Mandlekar et al., 2021), and 10 tasks from Metaworld Yu et al. (2020). On Metaworld, WARPD achieves comparable performance to Diffusion Policy but with a $\sim 45x$ reduction in FLOPs per step, representing a significant improvement in computational efficiency (FLOPs per step are the floating point operations, amortized over all steps of the episode). Analysis across a range of benchmark robotic locomotion and manipulation tasks, demonstrates WARPD's ability to accurately capture the *behavior distribution* of diverse trajectories, showcasing its capacity to learn a distribution of behaviors.

## 2 RELATED WORK

### 2.1 IMITATION LEARNING AND DIFFUSION FOR ROBOTICS

Behavioral cloning has progressed with transformer-based models such as PerAct (Shridhar et al., 2022) and RT-1 (Brohan et al., 2022), which achieve strong task performance. Vision-language models like RT-2 (Brohan et al., 2023) interpret actions as tokens, while RT-X (Collaboration et al., 2023) generalizes across robot embodiments. Object-aware representations (Heravi et al., 2022), energy-based models, and temporal abstraction methods (implicit behavioral cloning (Florence et al., 2022), sequence compression (Zheng et al., 2024)) improve multitask learning. DBC (Chen et al., 2024) increases robustness to sensor noise (this is complementary to WARPD, which targets dynamics perturbations such as object shifts or execution-time disturbances). Diffusion models, originally introduced for generative modeling (Ho et al., 2020a; Rombach et al., 2022a), have become powerful tools for robotics. Trajectory-based approaches capture multimodal action distributions (Chi et al., 2024), while goal-conditioned methods such as BESO (Reuss et al., 2023) and Latent Diffusion Planning (Kong et al., 2024) improve efficiency through latent conditioning. Diffusion has

also been applied to grasping and motion planning (Urain et al., 2022; Luo et al., 2024; Carvalho et al.), skill chaining (Mishra et al., 2023), and locomotion (Huang et al., 2024). Hierarchical extensions including ChainedDiffuser (Xian & Gkanatsios, 2023), SkillDiffuser (Liang et al., 2024b), and multitask latent diffusion (Tan et al., 2024) address long-horizon planning. Recently, OCTO (Octo Model Team et al., 2024) demonstrates diffusion-based generalist robot policies. RDP (Xue et al., 2025) performs diffusion in latent action chunk space to speed up inference.

## 2.2 HYPERNETWORKS AND POLICY GENERATION

Hypernetworks, introduced by Ha et al. (2016), generate parameters for secondary networks and have been applied in multiple domains. They were first used for meta-learning in one-shot learning tasks (Bertinetto et al., 2016) and more recently extended to robot policy representations (Hegde et al., 2024). This direction aligns with Dynamic Filter Networks (Jia et al., 2016), which emphasize adaptability to input data. Latent Diffusion Models (LDMs) have also been used to model training dynamics in parameter spaces (Peebles et al., 2022). LDMs have enabled behavior-conditioned policies from text (Hegde et al., 2023) and trajectory embeddings (Liang et al., 2024a), as well as architectures distributions such as ResNets (Wang et al., 2024). Unlike Hegde et al. (2023) and Liang et al. (2024a), which rely on pre-collected policy datasets, WARPD requires a dataset of trajectories.

## 2.3 WORLD MODELS

Ha & Schmidhuber (2018) introduced world models for forecasting in latent space. PlaNet (Hafner et al., 2019b) added pixel-based dynamics learning and online planning. Dreamer (Hafner et al., 2019a) learned latent world models with actor-critic RL for long horizons, followed by DreamerV2 (Hafner et al., 2020) with discrete representations achieving human-level Atari, and DreamerV3 (Hafner et al., 2023) scaling across domains. IRIS (Micheli et al., 2023) applied transformers for sequence modeling, reaching superhuman Atari in two hours. SLAC (Lee et al., 2019) showed stochastic latent variables accelerate RL from high-dimensional inputs. VINs (Tamar et al., 2016) embedded differentiable value iteration for explicit planning, while E2C (Watter et al., 2015) combined VAEs with locally linear dynamics. DayDreamer (Wu et al., 2022) enabled real robot learning in one hour, and MILE (Hu et al., 2022) adapted Dreamer to CARLA with $31\%$ gains. Popov et al. (2024) scaled model-based imitation learning to large self-driving datasets. Recent work includes SafeDreamer (Zhang et al., 2023) for safety, STORM (Micheli et al., 2024) with efficient transformers, UniZero (Zhang et al., 2024) for joint model-policy optimization, and Time-Aware World Models (Chen et al., 2025) capturing temporal dynamics. Beyond these, large-scale pretraining and multimodal foundations extend world models. V-JEPA 2 (Assran et al., 2025) demonstrated self-supervised video models. DINO-based methods, including Back to the Features (Baldassarre et al., 2025) and DINO-WM (Zhou et al., 2024), leverage pre-trained visual features. NVIDIA's Cosmos platform (NVIDIA et al., 2025) proposes a foundation model ecosystem for physical AI. Vid2World (Huang et al., 2025) adapts video diffusion models to interactive world modeling, and Pandora (Xiang et al., 2024) integrates natural language actions with video states.

## 3 METHOD & PROBLEM FORMULATION

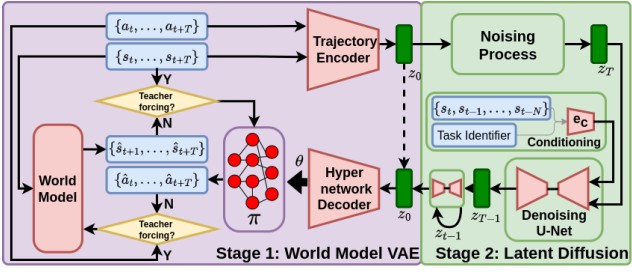

Figure 2: **WARPD**: Stage 1: Pre-train a VAE and world model. The VAE encodes trajectories into a latent space and decodes them as policy parameters, which are optimized for behavior cloning and trajectory tracking With teacher forcing enabled, the world model is optimized; when disabled, it optimizes the VAE. Stage 2: Train a conditional latent diffusion model to learn the latent distribution.

We address policy neural network weight generation, inspired by Hegde et al. (2023), which used latent diffusion to model policy parameter distributions but relied on policy datasets that are often unavailable. Our method, WARPD, instead trains on trajectory datasets through a two-step process: a variational autoencoder (VAE) with weak KL regularization encodes trajectories into a latent space, decoded by a conditioned hypernetwork into policy weights optimized with a co-trained world model. During "teacher forcing", the world model is trained to model the state transitions using ground truth data.

We use this trained world model to guide the generated policy to always be in the desired trajectory state distribution. Then, a diffusion model learns the latent distribution (see fig. 2).

Compared to Hegde et al. (2023), which encodes policy parameters and employs a graph hypernetwork with a MSE loss on parameter reconstruction, our approach differs as it: (1) encodes trajectories as opposed to parameters, into latent space (i.e., we do not require a dataset of policies) (2) uses a simple hypernetwork, (3) applies a behavior cloning loss (detailed in section 3.3 & section 3.2) on the generated policy, and (4) learns a world model for predicting observations given the action in an environment. Below we discuss the problem formulation and derivation.

### 3.1 LATENT POLICY REPRESENTATION

We begin by formulating our approach for unconditional policy generation. Assume a distribution over stochastic policies, where variability reflects behavioral diversity. Each policy is parameterized by $\theta$, with $\pi(\cdot, \theta)$ denoting a sampled policy and $p(\theta)$ the parameter distribution. Sampling a policy corresponds to drawing $\theta \sim p(\theta)$. When a policy interacts with the environment, it gives us a trajectory $\tau = \{s_t, a_t\}_{t=0}^T$. We assume multiple such trajectories are collected by repeatedly sampling $\theta$ and executing the corresponding policy. This enables a heterogeneous dataset, e.g., from humans or expert agents. For a given $\theta$, actions are noisy: $a_t \sim \mathcal{N}(\pi(s_t, \theta), \sigma^2)$.

Our objective is to recover the distribution $p(\theta)$ that generated the trajectory dataset. We posit a latent variable $z$ capturing behavioral modes, and assume conditional independence: $p(\tau \mid z, \theta) = p(\tau \mid \theta)$. Given trajectory data, we maximize the likelihood $\log p(\tau)$. To do so, we derive a modified Evidence Lower Bound (mELBO) that incorporates $p(\theta)$ (see below).

$$\log p(\tau) = \log \int \int p(\tau, \theta, z)\, dz\, d\theta \quad \text{(Introduce policy parameter } \theta \text{ and latent variable } z\text{)}$$

$$= \log \int \int p(\tau \mid z, \theta) p(\theta \mid z) p(z)\, dz\, d\theta \quad \text{(Apply the chain rule)}$$

$$= \log \int \int \frac{p(\tau \mid z, \theta) p(\theta \mid z) p(z)}{q(z \mid \tau)} q(z \mid \tau)\, dz\, d\theta \tag{1a}$$

$$\text{(Introduce a variational distribution } q(z \mid \tau)\text{, approximating the true posterior } p(z \mid \tau)\text{)}$$

$$= \log \int \mathbb{E}_{p(\theta \mid z)} \left[ \frac{p(\tau \mid z, \theta) p(z)}{q(z \mid \tau)} q(z \mid \tau) \right] dz \tag{1b}$$

$$\geq \mathbb{E}_{q(z \mid \tau)} \left[ \log \left( \frac{\mathbb{E}_{p(\theta \mid z)} \left[ p(\tau \mid z, \theta) \right] p(z)}{q(z \mid \tau)} \right) \right] \quad \text{(Jensen's inequality)}$$

$$= \mathbb{E}_{q(z \mid \tau)} \left[ \log \left( \mathbb{E}_{p(\theta \mid z)} \left[ p(\tau \mid z, \theta) \right] \right) \right] - \mathbb{E}_{q(z \mid \tau)} \left[ \log \left( q(z \mid \tau) \right) - \log \left( p(z) \right) \right] \tag{1c}$$

$$= \mathbb{E}_{q(z \mid \tau)} \left[ \log \left( \mathbb{E}_{p(\theta \mid z)} \left[ p(\tau \mid \theta) \right] \right) \right] - \mathrm{KL}(q(z \mid \tau) \,\|\, p(z)) \quad \text{(cond. independence)} \tag{1d}$$

$$\geq \mathbb{E}_{q(z \mid \tau)} \left[ \mathbb{E}_{p(\theta \mid z)} \left[ \log \left( p(\tau \mid \theta) \right) \right] \right] - \mathrm{KL}(q(z \mid \tau) \,\|\, p(z)) \quad \text{(Jensen's inequality)} \tag{1e}$$

Assuming the state transitions are Markov and $s_1$ is independent of $\theta$, the joint likelihood of the entire sequence $\{(s_1, a_1), (s_2, a_2), \ldots, (s_T, a_T)\}$ (i.e., $p(\tau \mid \theta)$) is given by:

$$p(s_1, a_1, \ldots, s_T, a_T \mid \theta) = p(s_1) p(a_1 \mid s_1, \theta) \cdot \prod_{t=2}^{T} p(s_t \mid s_{t-1}, a_{t-1}) p(a_t \mid s_t, \theta) \tag{2a}$$

$$\log p(s_1, a_1, \ldots, s_T, a_T \mid \theta) = \log p(s_1) + \log p(a_1 \mid s_1, \theta)$$

$$+ \sum_{t=2}^{T} \left[ \log p(s_t \mid s_{t-1}, a_{t-1}) + \log p(a_t \mid s_t, \theta) \right] \tag{2b}$$

Substituting $2b$ in $1e$:

$$\log p(\tau) \geq \mathbb{E}_{q(z \mid \tau)} \left[ \mathbb{E}_{p(\theta \mid z)} \left[ \log \left( p(\tau \mid \theta) \right) \right] \right] - \mathrm{KL}(q(z \mid \tau) \,\|\, p(z))$$

$$= \mathbb{E}_{q(z \mid \tau)} \left[ \mathbb{E}_{p(\theta \mid z)} \left[ \sum_{t=1}^{T} \log p(a_t \mid s_t, \theta) + \sum_{t=2}^{T} \log p(s_t \mid s_{t-1}, a_{t-1}) \right] \right]$$

$$- \mathrm{KL}(q(z \mid \tau) \,\|\, p(z)) + A \tag{3}$$

Where $A$ consists of $\log p(s_1)$, and since this cannot be subject to maximization, we shall ignore it.

Therefore, our modified ELBO is:

$$\mathbb{E}_{q(z|\tau)}\left[\mathbb{E}_{p(\theta|z)}\left[\sum_{t=1}^{T}\underbrace{\log p(a_t \mid s_t, \theta)}_{Behavior\ Cloning} + \sum_{t=2}^{T}\underbrace{\log p(s_t \mid s_{t-1}, a_{t-1})}_{World\ Model}\right]\right] - \underbrace{\text{KL}(q(z \mid \tau) \parallel p(z))}_{KL\ Regularizer} \quad (4)$$

### 3.2 Loss function

Since we now have a modified ELBO objective, we shall now try to approximate its components with a variational autoencoder and a world model. Let $\phi_{enc}$ be the parameters of the VAE encoder that variationally maps trajectories to $z$, $\phi_{dec}$ be the parameters of the VAE decoder, and $\phi_{wm}$ be the world model parameters. We assume the latent $z$ is distributed with mean zero and unit variance. We construct the VAE decoder to approximate $p(\theta \mid z)$ with $p_{\phi_{dec}}(\theta \mid z)$. Considering $a_t \sim \mathcal{N}(\pi(s_t, \theta), \sigma^2)$, and $\tau_k = \{s_t^k, a_t^k\}_{t=1}^{T}$, we derive our VAE loss function as:

$$\mathcal{L}_{BC} = \sum_{t=1}^{T}\mathbb{E}_{q_{\phi_{enc}}(z|\tau_k)}\left[(a_t^k - \pi(s_t^k, f_{\phi_{dec}}(z)))^2\right]$$

$$\mathcal{L}_{RO} = \sum_{t=2}^{T}\mathbb{E}_{q_{\phi_{enc}}(z|\tau_k)}\left[\text{KL}\big(p_{\phi_{wm}}(s_t \mid s_{t-1}^k, \pi(s_{t-1}^k, f_{\phi_{dec}}(z))) \parallel p_{\phi_{wm}}(s_t \mid s_{t-1}^k, a_{t-1}^k))\right]$$

$$\mathcal{L}_{TF} = \sum_{t=2}^{T}(s_t^k - \hat{s}_t^k)^2 \qquad \mathcal{L}_{KL} = \beta_{kl}\sum_{i=1}^{\dim(z)}\left(\sigma_{e_i}^2 + \mu_{e_i}^2 - 1 - \log \sigma_{e_i}^2\right)$$

$$\mathcal{L}\left(\{s_t^k, a_t^k\}_{t=1}^{T} \mid \phi_{enc}, \phi_{dec}, \phi_{wm}\right) = \mathcal{L}_{BC} + \mathcal{L}_{RO} + \mathcal{L}_{TF} + \mathcal{L}_{KL} \quad (5)$$

where, $\mathcal{L}_{BC}$ is the behavior cloning loss to train the policy decoder, $\mathcal{L}_{RO}$ is the rollout loss to correct the decoded policy's actions using the world model, $\mathcal{L}_{TF}$ is the teacher forcing loss to train the world model, and $\mathcal{L}_{KL}$ is the KL loss to regularize the latent space. $\theta$ is obtained from the hypernetwork decoder $f_{\phi_{dec}}(z)$. $(\mu_e, \sigma_e) = f_{\phi_{enc}}(\{s_t^k, a_t^k\}_{t=1}^{T})$, $z \sim \mathcal{N}(\mu_e, \sigma_e)$, $\hat{s}_t^k \sim p_{\phi_{wm}}(s_t^k \mid s_{t-1}^k, a_{t-1}^k)$ and $\beta_{kl}$ is the regularization weight. The complete derivation is shown in section A.1. Since the decoder in the VAE outputs the parameter of a secondary network, we shall use a conditional hypernetwork, specifically the model developed for continual learning by (von Oswald et al., 2020). For computational stability, we shall use $\mathcal{L}_{BC}$, $\mathcal{L}_{RO}$ and $\mathcal{L}_{KL}$ to optimize the VAE (encoder and decoder parameters) and $\mathcal{L}_{TF}$ to train the world model parameters. With the teacher forcing objective we get a reliable world model that we can then use in the rollout objective. This is similar to procedures followed in Assran et al. (2025); Popov et al. (2024); Hu et al. (2022). In practice, we see that approximating $p(z) = \mathcal{N}(0, I)$ is suboptimal, and therefore we set $\beta_{kl}$ to a very small number $\sim (10^{-10}, 10^{-6})$. After training the VAE to maximize the objective provided in eq. (5) with this $\beta_{kl}$, we have access to this latent space $z$ and can train a diffusion model to learn its distribution $p(z)$. We can condition the latent denoising process on the current state and/or the task identifier $c$ of the policy required. Therefore the model shall be approximating $p_{\phi_{dif}}(z_{t-1} \mid z_t, c)$. After denoising for a given state and task identifier, we can convert the denoised latent to the required policy. Therefore, to sample from $p(\theta)$, first sample z using the trained diffusion model $z \sim p_{\phi_{dif}}(z_0)$, and then apply the deterministic function $f_{\phi_{dec}}$ to the sampled $z$. Note that to sample policies during inference, we do not need to encode trajectories; rather, we need to sample a latent using the diffusion model and use the hypernetwork decoder of a pre-trained VAE to decode a policy from it.

### 3.3 Positioning to Prior Work

In table 1, we compare WARPD with closely related methods. While many other methods have conceptual overlap with our method, WARPD is the only method that uses diffusion to generate policy parameters with trajectory datasets (without any reward data). Further, we use model-based imitation learning with world models to further guide our generated policies. The necessity of the components used used in WARPD is based on the derivation described in section 3.3.

| Method | Diffusion based | Generates policy params (not traj.) | Trajectory data only (no reward signal) | World model |
|---|---|---|---|---|
| WARPD (ours) | ✓ | ✓ | ✓ | ✓ |
| Chi et al. (2024) | ✓ | ✗ | ✓ | ✗ |
| Xue et al. (2025) | ✓ | ✗ | ✓ | ✗ |
| Hegde et al. (2023) | ✓ | ✓ | ✗ | ✗ |
| Liang et al. (2024a) | ✓ | ✓ | ✗ | ✗ |
| Zhu et al. (2025) | ✓ | ✗ | ✗ | ✓ |
| Hegde et al. (2024) | ✗ | ✓ | ✗ | ✗ |
| Pu et al. | ✗ | ✗ | ✗ | ✓ |
| Hafner et al. (2023) | ✗ | ✗ | ✗ | ✓ |
| Zhang et al. (2024) | ✗ | ✗ | ✗ | ✓ |
| Hu et al. (2022) | ✗ | ✗ | ✓ | ✓ |
| Popov et al. (2024) | ✗ | ✗ | ✓ | ✓ |

Table 1: Comparison of WARPD (ours) with closely related work. WARPD is a conceptually novel framework for generating policy parameters with state-action only trajectory datasets.

## 4 EXPERIMENTS

We run four sets of experiments. In the first set (section 4.1), we evaluate the validity of our main contributions. In the second set (section 4.2), we ablate different components of our method. In the third set (section 4.3), we show how WARPD can be scaled to vision-based observation environments. In the final set (section 4.4), we analyze the behavior distribution modeled by our latent space. In the first set, we compare WARPD with action trajectory generation methods with respect to 1) Longer Action Horizons and Environment Perturbations, where experiments are performed while varying these parameters on the PushT task (Chi et al., 2024) and the Lift and Can Robomimic tasks (Mandlekar et al., 2021), and 2) Lower inference costs, where experiments are performed on 10 tasks from the Metaworld Yu et al. (2020) suite of tasks, to show WARPD requires fewer parameters during inference while maintaining multi-task performance. The task descriptions are provided in section A.5. We choose a multi-task experiment here as the model capacity required for solving multiple tasks generally increases with the number of tasks.

We focus on demonstrating results in state-based observation spaces. Our generated policies are Multi-Layer Perceptrons (MLP) with 2 hidden layers with 256 neurons each. In the VAE, the encoder is a sequential network that flattens the trajectory and compresses it to a low-dimensional latent space, and the decoder is a conditional hypernetwork (Ehret et al., 2021). The details of the VAE implementation are provided in section A.8.2 and section A.8.3. For the world model, since we use low-dimensional observation spaces, we use a simple MLP with 2 hidden layers with 1024 neurons each to map the history of observations and actions to the next observation. For stability, we use $\mathcal{L}_{RO}$ only after 10 epochs of training. This warm-starts the world model before we use it to optimize the policy generator. For all experiments, the latent space is $\mathbb{R}^{256}$ and the learning rate is $10^{-4}$ with the Adam optimizer. For the diffusion model, we use the DDPM Scheduler for denoising. Based on the results are shown in section A.3 (inspired by Chi et al. (2024)), we chose the ConditionalUnet1D model for all experiments in the paper. Just as Chi et al. (2024), we condition the diffusion model with FiLM layers, and also use the Exponential Moving Average (He et al., 2020) of parameter weights (commonly used in DDPM) for stability. All results presented are obtained over three seeds, and the compute resources are described in section A.9

### 4.1 EMPIRICAL EVALUATION OF CONTRIBUTIONS

#### 4.1.1 LONGER ACTION HORIZONS & ROBUSTNESS TO PERTURBATIONS

We first evaluate our method on the PushT task (Chi et al., 2024), a standard benchmark for diffusion-based trajectory generation in manipulation. The goal is to align a 'T' block with a target position and orientation on a 2D surface. Observations consist of the end-effector's position and the block's position and orientation. Actions specify the end-effector's target position at each time step. Success rate is defined as the maximum overlap between the actual and desired block poses during a rollout. We test under different action horizons and varying levels of environment perturbation, simulated via an adversarial agent that randomly displaces the 'T' block.

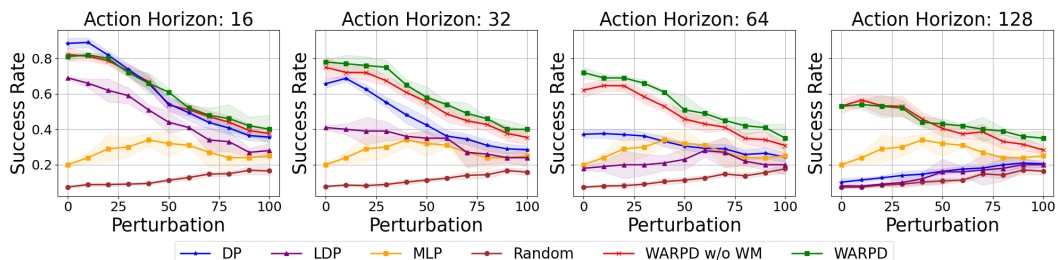

Figure 3: **Longer action horizons and robustness to perturbations on PushT**: Performance of WARPD and baselines on the PushT task on variable action horizon and environment perturbations.

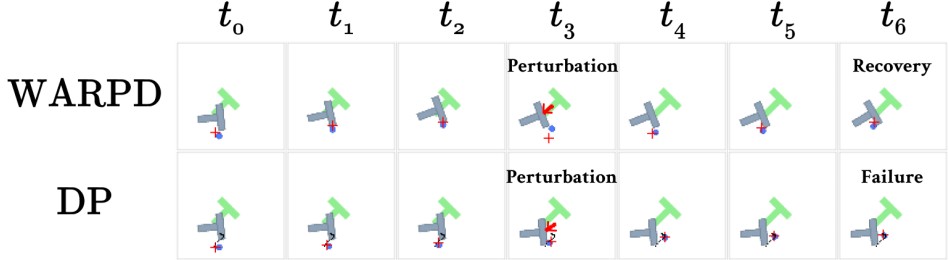

Figure 4: **Visualization of Perturbation**: When an adversarial perturbation is applied, we see that WARPD's generated closed-loop policy successfully adapts to the change.

For the WARPD model, we first train a VAE to encode trajectory snippets (of length equal to the action horizon) into latents representing locally optimal policies. These policies are optimized with a co-trained world model. A conditional latent diffusion model, given the current state, then generates a latent that the VAE decoder transforms into a locally optimal policy for the next action horizon. The inference process is illustrated in fig. 1. We train two variants of WARPD, with (WARPD) and without (WARPD w/o WM) the world model (i.e., we train WARPD with just $\mathcal{L}_{BC} + \mathcal{L}_{KL}$).

As baselines for this experiment, we compare the proposed WARPD variants against four alternatives: 1) a **Diffusion Policy (DP)** model that generates open-loop action trajectories for a fixed action horizon; 2) a **Latent Diffusion Policy (LDP)** model, which is structurally similar to WARPD but decodes the latent representation into an action trajectory rather than a closed-loop policy; 3) a **Multilayer Perceptron (MLP)** policy, which shares the same architecture as the policy network generated by WARPD and serves to isolate the impact of diffusion modeling; 4) a **Random Policy**, which provides a lower-bound performance reference. For a fair comparison, all diffusion-based models (WARPD, DP, and LDP) use the same diffusion model size and hyperparameters, corresponding to the `medium` configuration described in section A.8.4 and section A.8.7. LDP uses a VAE decoder, implemented as an MLP with two hidden layers of 256 neurons each, to output an action chunk of the same length as the action horizon.

All models are evaluated across 50 uniquely seeded environment instances, with each evaluation repeated 10 times, across 3 training seeds. Figure 3 illustrates the impact of perturbation magnitudes and action horizons on success rates across all baselines. Perturbations refer to random displacements applied to the `T` block, occurring at randomly selected time steps with 10% probability. A sample rollout with a perturbation magnitude of 50 is shown in fig. 4.

While DP demonstrates comparable performance to both WARPD variants at an action horizon of 16 with minimal perturbations, WARPD exhibits superior robustness as the action horizon increases. This enhanced robustness of WARPD with the world model becomes more pronounced in the presence of larger perturbations. Specifically, at longer action horizons such as 128, WARPD w/ WM maintains a significantly higher success rate compared to DP across all perturbation levels. The MLP generally underperforms compared to both WARPD variants and DP, highlighting the benefits of diffusion-based approaches for this task. LDP has a lower success rate than WARPD, indicating that generating a closed-loop policy is more important than learning the latent representation space. The relatively lower sensitivity to perturbations at an action horizon of 16 for both policies can be attributed to the more frequent action trajectory queries inherent in DP at shorter horizons (i.e. smaller action chunks), effectively approximating a more closed-loop control strategy.

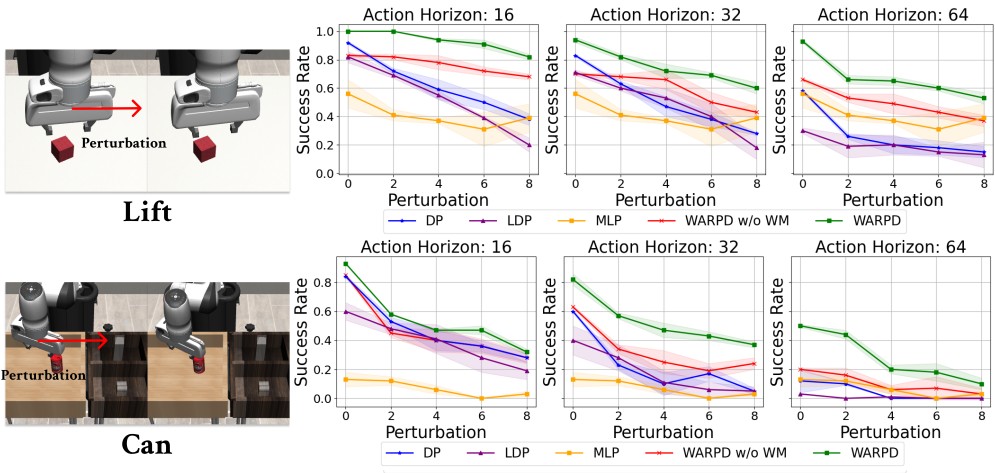

Figure 5: **Longer action horizons and robustness to perturbations on Robomimic tasks**: Performance of WARPD and DP as we vary the action horizon and environment perturbations.

We also ran experiments on the Robomimic (Mandlekar et al., 2021) Lift and Can tasks, using the same hyperparameters as the PushT experiment, the same task settings, and the mh demonstration data from (Chi et al., 2024). To simulate perturbations, we add random translation and rotation vectors to the end effector, applied 10% of the time. fig. 5 shows the performance of the WARPD variants and baselines under these perturbations across different action horizons. The x-axis corresponds to perturbation magnitude. Similar to PushT, WARPD outperforms DP for longer horizons and is more robust to perturbations. Here, we see that WARPD also significantly outperforms WARPD w/o WM. We believe that this is because the state density of the provided dataset is higher in PushT as compared to Robomimic, and model-based imitation learning (with the world model) provides robustness to covariate shift (Popov et al., 2024; Hu et al., 2022).

### 4.1.2 LOW INFERENCE COST

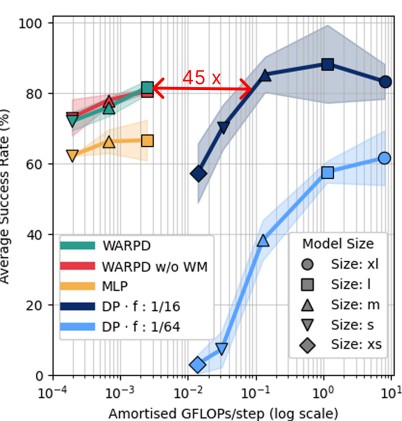

Figure 6: **Success rate vs. average compute** of WARPD, DP, and MLP policies on 10 Metaworld tasks for various model sizes. The x-axis shows the GFLOPs/step for each policy on a log scale. WARPD performs ∼ 45x fewer inference computations than a DP policy with comparable performance.

We will now look at the next contribution, namely, lower inference cost compared to methods that diffuse action trajectories instead of policies. When training a single policy on multiple tasks, it is known that a larger model capacity is needed. This is detrimental in robotics applications as this increases control latency. We train a task-conditioned WARPD model and show that the cost of task generalization is borne by the latent diffusion model, **while the generated execution policy remains small**. Because WARPD generates a smaller policy, the runtime compute required for inference is lower than SOTA diffusion methods.

We experiment on 10 tasks of the Metaworld benchmark, the details of which are in section A.5. We set the action horizon to the length of the entire trajectory for WARPD to generate policies that shall work for the entire duration of the rollout, where at each time step, the generated MLPs shall predict instantaneous control. We experimented over three sizes of the generated MLP policy: 128, 256, and 512 neurons per layer, each having 2 hidden layers. We also train 10 DP models, spread over a grid of 5 different sizes (xs, s, m, l, xl) and 2 action horizons: 32 and 128. Each DP model is run at an inference frequency of half the action horizon. We provide the details of the DP model in section A.8.1. Finally, we also train 3 MLP models with 128, 256, and 512 neurons per layer, as baselines.

| Component/ Claim | Ablation / Baseline | What It Tests | Where in Paper |
|---|---|---|---|
| With and without a World model | WARPD w/o WM | Does modeling dynamics and using rollout loss actually help? | section 4.1.1 |
| Policy-space vs action-space diffusion | Diffusion Policy (DP) | Is diffusing actions sufficient vs generating policies? | section 4.1.1 |
| Policy-space vs latent trajectory | Latent Diffusion Policy (LDP) | Is decoding latent action chunks comparable to decoding weights? | section 4.1.1 |
| Need for policy generation at all | MLP policy (no diffusion) | Does simple BC on the same architecture suffice? | section 4.1.1; section 4.1.2 |
| Strength of KL regularization ($\beta$) | $\beta$ sweep ($10^{-7}$, $10^{-9}$, $10^{-10}$) | Does strong $\mathcal{N}(0, I)$ regularization help or hurt? | section A.4.1 |

Table 2: Summary of key components and the corresponding ablations or baselines that test them. More ablations are analyzed in section 4.2.

Note that WARPD uses a fixed action horizon equal to the full episode length (500 steps), whereas the DP model uses a variable horizon. The WARPD inference process is illustrated on the right-hand side of fig. 1. All baseline models receive the task identifier as part of the state input. Each model is trained with 3 random seeds, and evaluated across 10 tasks, with 16 rollouts per task. fig. 6 presents the results of this evaluation. In the plot, the x-axis represents average per-step inference compute (in GFLOPs), and the y-axis indicates the overall success rate across tasks. For DP models, achieving high success rates requires increasing model size or denoising frequency (i.e., predicting shorter action chunks), both of which raise computational cost. In contrast, WARPD generates a simpler, more efficient controller, requiring significantly less compute. The best-performing WARPD model achieves an 81% success rate with $\sim 45\times$ fewer inference operations than the closest-performing DP model. Interestingly, the MLP baseline also performs well, and is comparable in efficiency to WARPD, but still lags in performance. We attribute this to the unimodal nature of this dataset, as MLPs struggled with the multimodal PushT task in the previous section. Note that the WARPD performed comparably to the w/o WM variant. In different scenarios, such as the state-conditioned experiments where the policy is regenerated more frequently, the generation cost could also be amortized. Even in such a conservative setting, when we incorporate the computational cost for generation (0.0227 GFLOPs), WARPD still requires $\sim 4.5\times$ fewer inference operations.

## 4.2 ABLATIONS

Considering that WARPD consists of multiple components, we analyze each one. We perform ablations over three components of our method: 1) Diffusion model architecture, section A.3; 2) VAE decoder size, section A.4; 3) KL coefficient for the VAE, section A.4.1. We find that: 1) a UNET converges faster than a transformer, 2) using a larger hypernetwork decoder increases the performance, 3) using a lower KL coefficient generates policies that better track a desired trajectory. Further, in section 4.1.1, we ablate the world model and see that it helps more in the Robomimic tasks than in the PushT task. We believe this is because the state space is more complex in Robomimic than that in PushT, whilst the number of trajectories remains roughly the same. This results in insufficient trajectories covering the state space, rendering the learned policy susceptible to covariate shift.

Since WARPD relies on several interacting components derived from our probabilistic formulation, we summarize their roles in table 2. The ablation results show that each theoretically motivated component is also empirically necessary, jointly justifying the overall design.

## 4.3 VISION OBSERVATION SCALING

We conducted initial experiments on the PushT image environment to evaluate the applicability of our method in vision-based tasks. Our approach involved pre-training a vision encoder to map images of the PushT environment to their corresponding ground truth states. We then trained WARPD to utilize these image embeddings as states. For comparison, we also trained a Diffusion Policy (DP) model on the same embeddings. The results for an action horizon of 64 are presented below.

As shown in table 3, WARPD consistently outperforms DP in the presence of increasing perturbation, demonstrating its robustness even when operating on image-derived state embeddings. These experiments strongly suggest that if an effective image embedding can be learned, the low-dimensional state space version of WARPD is readily applicable to vision-based tasks. This serves as an encouraging proof-of-concept for

| Perturbation | WARPD | DP |
|---|---|---|
| 0 | $0.54 \pm 0.05$ | $\mathbf{0.57 \pm 0.05}$ |
| 20 | $\mathbf{0.53 \pm 0.01}$ | $0.50 \pm 0.05$ |
| 40 | $\mathbf{0.45 \pm 0.01}$ | $0.42 \pm 0.05$ |
| 60 | $\mathbf{0.41 \pm 0.08}$ | $0.34 \pm 0.02$ |
| 80 | $\mathbf{0.36 \pm 0.06}$ | $0.30 \pm 0.02$ |
| 100 | $\mathbf{0.28 \pm 0.05}$ | $0.24 \pm 0.06$ |

Table 3: **PushT Image results** with horizon 64

WARPD's generalizability beyond state-based environments. It can be noted here as well that a diffusion model's inference cost ($\sim 3.99$ GFLOPs) is still much greater than the hypernetwork decoder ($\sim 0.056$ GFLOPs) and the ResNet18 vision encoder ($\sim 0.334$ GFLOPs)

### 4.4 BEHAVIOR ANALYSIS

WARPD models trajectory data from a distribution of policies, exposing this distribution through its latent space. On the Robomimic Lift task with the MH dataset (300 trajectories from 6 operators of varied proficiency: 2 "worse," 2 "okay," and 2 "better."), WARPD encoded entire demonstration trajectories. A 2D t-SNE plot revealed clusters aligned with operator identity, despite WARPD receiving no explicit operator labels. This shows WARPD can cluster behaviors and potentially filter unwanted ones. This is further studied in section A.7.

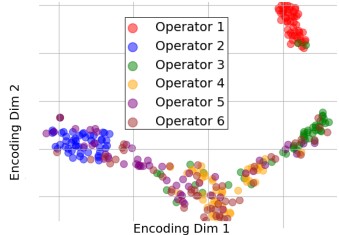

Figure 7: **Behavior distribution**

## 5 LIMITATIONS AND FUTURE WORK

While WARPD is a promising framework for policy generation, Diffusion Policy (DP) performs better in short-horizon, low-perturbation settings. This gap likely stems from VAE approximation errors and WARPD's added complexity. Another limitation is the additional training overhead compared to traditional diffusion policy models (see section A.9). The world model is a key component when covariate shift is significant, as illustrated by the performance gap between WARPD and WARPD w/o WM on Robomimic. At the same time, the behavior cloning loss ensures that, in the limit of a weak or undertrained world model, WARPD behaves similarly to a diffusion-augmented BC model rather than failing catastrophically. Compared to standard trajectory-diffusion policies, our training pipeline introduces additional overhead (VAE + world model + diffusion), which we detail in section A.9; this is comparable to other world model-based imitation learning methods. Our primary target is regimes where training is offline but runtime compute is constrained, and in this setting, WARPD offers substantial FLOPs-per-step savings while maintaining or improving performance.

Thus, future work could improve WARPD's VAE decoder through chunked deconvolutional hypernetworks (von Oswald et al., 2020), enabling more efficient decoding. Extending WARPD to Transformer or ViT policies is another direction, especially for sequential or visual tasks (Dosovitskiy et al., 2020). Incorporating WARPD to foundation VLA models as an action head is another exciting avenue. Finally, warm-starting with prior latents (Chi et al., 2024) may further boost performance by providing richer priors.

## 6 CONCLUSION

We introduce World Model Assisted Reactive Policy Diffusion (WARPD), a novel framework for learning a distribution over policies from diverse demonstration trajectories. WARPD models behavioral diversity via latent diffusion, a world model, and uses a hypernetwork decoder to generate policy weights, enabling closed-loop control directly from sampled latents. Our evaluation highlights two key strengths of WARPD: robustness and computational efficiency. Compared to Diffusion Policy, WARPD delivers more reliable performance in environments with long action horizons and perturbations, while reducing inference costs, especially in multi-task settings.

## REPRODUCIBILITY STATEMENT

All reported results are averaged over three random seeds to ensure statistical reliability. We provide full implementation details of model architectures, training objectives, hyperparameters, and evaluation protocols in the main text and appendix.

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

## A  APPENDIX

### A.1  VAE LOSS DERIVATION

Since $a_t \sim \mathcal{N}(\pi(s_t, \theta), \sigma^2)$:

$$p(a_t \mid s_t, \theta) = \frac{1}{\sqrt{2\pi\sigma^2}} \exp\left(-\frac{(a_t - \pi(s_t, \theta))^2}{2\sigma^2}\right) \tag{6}$$

Our objective is to maximize the $mELBO$. The negative log likelihood of trajectory $\tau_k = \{s_t^k, a_t^k\}_{t=1}^T$ for the given VAE parameters is:

$$\mathcal{L}\left(\tau_k \mid \phi_{enc}, \phi_{dec}, \phi_{wm}\right)$$

$$= -\sum_{t=1}^T \mathbb{E}_{q_{\phi_{enc}}(z|\tau_k)}\left[\mathbb{E}_{p_{\phi_{dec}}(\theta|z)}\left[\log p\left(a_t^k \mid s_t^k, \theta\right)\right]\right]$$

$$- \sum_{t=2}^T \mathbb{E}_{q_{\phi_{enc}}(z|\tau_k)}\left[\mathbb{E}_{p_{\phi_{dec}}(\theta|z)}\left[\log p_{\phi_{wm}}(s_t^k \mid s_{t-1}^k, a_{t-1}^k)\right]\right]$$

$$+ \mathrm{KL}\left(q_{\phi_{enc}}\left(z \mid \tau_k\right) \| p(z)\right) \tag{7}$$

Consider the second term in the above equation. On maximization $a_{t-1}^k = \pi(s_{t-1}^k, \theta)$, and because the inner quantity is a constant w.r.t. $s_t$ we can add a harmless expectation $\mathbb{E}_{s_t \sim \pi}[.]$ (i.e., states visited by the estimated policy, not necessarily those in the dataset), therefore it becomes:

$$\mathbb{E}_{q_{\phi_{enc}}(z|\tau_k)}\left[\mathbb{E}_{p_{\phi_{dec}}(\theta|z)}\left[\mathbb{E}_{s_t \sim \pi}\left[\log p_{\phi_{wm}}(s_t^k \mid s_{t-1}^k, \pi(s_{t-1}^k, \theta))\right]\right]\right]$$

$$= \mathbb{E}_{q_{\phi_{enc}}(z|\tau_k)}\left[\mathbb{E}_{p_{\phi_{dec}}(\theta|z)}\left[\mathbb{E}_{s_t \sim \pi}\left[\log \frac{p_{\phi_{wm}}(s_t^k \mid s_{t-1}^k, \pi(s_{t-1}^k, \theta))}{p_{\phi_{wm}}(s_t^k \mid s_{t-1}^k, a_{t-1}^k)}\right]\right]\right]$$

$$+ \mathbb{E}_{q_{\phi_{enc}}(z|\tau_k)}\left[\mathbb{E}_{p_{\phi_{dec}}(\theta|z)}\left[\mathbb{E}_{s_t \sim \pi}\left[\log p_{\phi_{wm}}(s_t^k \mid s_{t-1}^k, a_{t-1}^k)\right]\right]\right] \tag{8}$$

We can now substitute in the KL term, and drop the expectation in the last term (since the inner terms only depend on $s_{t-1}^k$ and not $s_t \sim \pi$, $\theta$, or $z$). Therefore, the loss now becomes:

$$\mathcal{L}\left(\tau_k \mid \phi_{enc}, \phi_{dec}, \phi_{wm}\right)$$

$$= C + \frac{1}{2\sigma^2} \sum_{t=1}^T \mathbb{E}_{q_{\phi_{enc}}(z|\tau_k)}\left[\mathbb{E}_{p_{\phi_{dec}}(\theta|z)}\left[(a_t^k - \pi(s_t^k, \theta))^2\right]\right]$$

$$+ \sum_{t=2}^T \mathbb{E}_{q_{\phi_{enc}}(z|\tau_k)}\left[\mathbb{E}_{p_{\phi_{dec}}(\theta|z)}\left[\mathrm{KL}\left(p_{\phi_{wm}}(s_t^k \mid s_{t-1}^k, \pi(s_{t-1}^k, \theta)) \| p_{\phi_{wm}}(s_t^k \mid s_{t-1}^k, a_{t-1}^k)\right)\right]\right]$$

$$- \sum_{t=2}^T \log p_{\phi_{wm}}(s_t^k \mid s_{t-1}^k, a_{t-1}^k)$$

$$+ \mathrm{KL}\left(q_{\phi_{enc}}\left(z \mid \tau_k\right) \| p(z)\right) \tag{9}$$

For computational stability, we construct our decoder to be a deterministic function $f_{\phi_{dec}}$, i.e., $p_{\phi_{dec}}(\theta \mid z)$ becomes $\delta(\theta - f_{\phi_{dec}}(z))$. Further, if we have a trained world model, we can approximate $s_t^k$ with $s_t$ (i.e., direct model output samples) in the second term. This is done so that we can optimize the world model and policy correction separately with the teacher forcing and rollout

objectives (similar to that followed in Assran et al. (2025). Therefore:

$$\mathcal{L}\left(\tau_k \mid \phi_{enc}, \phi_{dec}, \phi_{wm}\right)$$

$$= C + \frac{1}{2\sigma^2} \sum_{t=1}^{T} \mathbb{E}_{q_{\phi_{enc}}(z|\tau_k)} \left[(a_t^k - \pi(s_t^k, f_{\phi_{dec}}(z)))^2\right]$$

$$+ \sum_{t=2}^{T} \mathbb{E}_{q_{\phi_{enc}}(z|\tau_k)} \left[\text{KL}\big(p_{\phi_{wm}}(s_t \mid s_{t-1}^k, \pi(s_{t-1}^k, f_{\phi_{dec}}(z))) \,\|\, p_{\phi_{wm}}(s_t \mid s_{t-1}^k, a_{t-1}^k))\right]$$

$$- \sum_{t=2}^{T} \log p_{\phi_{wm}}(s_t^k \mid s_{t-1}^k, a_{t-1}^k)$$

$$+ \text{KL}\left(q_{\phi_{enc}}(z \mid \tau_k) \,\|\, p(z)\right)$$

Where $C$ is a constant from the substitution. Enforcing $p(z) = \mathcal{N}(0, I)$, and ignoring constants, we get:

$$\mathcal{L}\left(\tau_k \mid \phi_{enc}, \phi_{dec}, \phi_{wm}\right) = \mathcal{L}_{BC} + \mathcal{L}_{RO} + \mathcal{L}_{TF} + \mathcal{L}_{KL} \tag{10}$$

$$\mathcal{L}_{BC} = \sum_{t=1}^{T} \mathbb{E}_{q_{\phi_{enc}}(z|\tau_k)} \left[(a_t^k - \pi(s_t^k, f_{\phi_{dec}}(z)))^2\right] \tag{11}$$

$$\mathcal{L}_{RO} = \sum_{t=2}^{T} \mathbb{E}_{q_{\phi_{enc}}(z|\tau_k)} \left[\text{KL}\big(p_{\phi_{wm}}(s_t \mid s_{t-1}^k, \pi(s_{t-1}^k, f_{\phi_{dec}}(z))) \,\|\, p_{\phi_{wm}}(s_t \mid s_{t-1}^k, a_{t-1}^k))\right]$$

$$\tag{12}$$

$$\mathcal{L}_{TF} = \sum_{t=2}^{T} (s_t^k - \hat{s}_t^k)^2 \tag{13}$$

$$\mathcal{L}_{KL} = \beta_{kl} \sum_{i=1}^{\dim(z)} \left(\sigma_{e_i}^2 + \mu_{e_i}^2 - 1 - \log \sigma_{e_i}^2\right) \tag{14}$$

where, $\mathcal{L}_{BC}$ is the behavior cloning loss to train the policy decoder, $\mathcal{L}_{RO}$ is the rollout loss to correct the decoded policy's actions using the world model, $\mathcal{L}_{TF}$ is the teacher forcing loss to train the world model, $\mathcal{L}_{KL}$ is the KL loss to regularize the latent space, $(\mu_e, \sigma_e) = f_{\phi_{enc}}(\tau_k)$, $z \sim \mathcal{N}(\mu_e, \sigma_e)$, $\hat{s}_t^k \sim p_{\phi_{wm}}(s_t^k \mid s_{t-1}^k, a_{t-1}^k)$ and $\beta_{kl}$ is the regularization weight.

## A.2    ABLATIONS

## A.3    DIFFUSION MODEL ARCHITECTURE

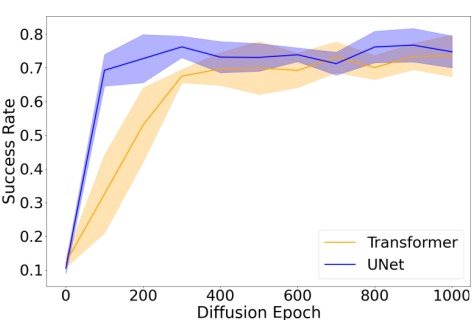

Figure 8: **Diffusion Architecture Ablation**

Diffusion models typically adopt either UNet-based Ho et al. (2020b) or Transformer-based Peebles & Xie (2023) architectures (described as medium "m" in section A.8.1). To guide our choice for the WARPD diffusion policy, we performed an ablation study on the PushT task (Chi et al., 2024) using an action horizon of 32. As shown in fig. 8, the UNet model demonstrated faster initial learning, achieving higher average success rates early in training. However, both architectures eventually converged to comparable final success rates. For consistency, we adopt the UNet architecture for all other experiments.

## A.4    DECODER SIZE

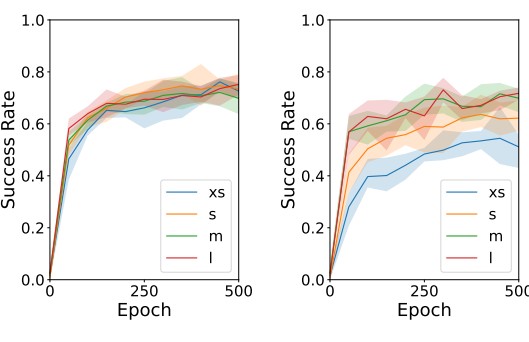

(a) Trajectory length 500     (b) Trajectory length 50

Figure 9: **Effect of VAE decoder size**: For long trajectories, even the smallest decoder ($xs$) yields high task performance, whereas short trajectories benefit from a larger decoder.

An interesting experiment was the effect of breaking a large trajectory into sub-trajectories and how this affects the latent space. A key takeaway from that experiment was that for halfcheetah locomotion, even small VAE decoders generated accurate policies from trajectory snippets. Whereas, for manipulation tasks from Metaworld, the same-sized small decoder was not capable of reconstructing the original policy. See section A.6 for this experiment. This finding prompted an ablation on the decoder size, evaluating the average success rate of decoded policies across all 10 Metaworld tasks. fig. 9 illustrates the performance of decoders with varying sizes, denoted as $xs$ (3.9M parameters), $s$ (7.8M parameters), $m$ (15.6M parameters), and $l$ (31.2M parameters). It's important to note that despite the substantial parameter count of the hypernetwork decoder, the resulting inferred policy remains relatively small ($< 100K$ parameters, see fig. 6). The results demonstrate that increasing the decoder size consistently improves the average success rate of the decoded policies. Refer section A.8.3 for more details regarding the decoder size characterization.

This contrasts with rollouts from the HalfCheetah environment, where even smaller decoders generated accurate policies from trajectory snippets. We hypothesize this discrepancy stems from two key factors. First, the cyclic nature of HalfCheetah provides sufficient information within snippets to infer the underlying policy. Second, the increased complexity of Metaworld tasks means that snippets may lack crucial information for inference. For instance, in a pick-and-place task, a snippet might only capture the "pick" action, leaving the latent without sufficient information to infer the "place" action.

### A.4.1    KL COEFFICIENT

A key hyperparameter in WARPD is the KL regularization term, $\beta_{\text{KL}}$, used during VAE training. In this section, we analyze its impact on the learned latent space using the PushT task with an action horizon of 32. We train three VAEs with $\beta_{\text{KL}}$ values of $1e-7$, $1e-9$, and $1e-10$. For evaluation, we sample a trajectory of length 32, encode and decode it via the VAE to generate a policy, and then execute this policy in the environment starting from the same initial state. We compute the MSE between the final state reached after 32 steps and the corresponding state in the original trajectory.

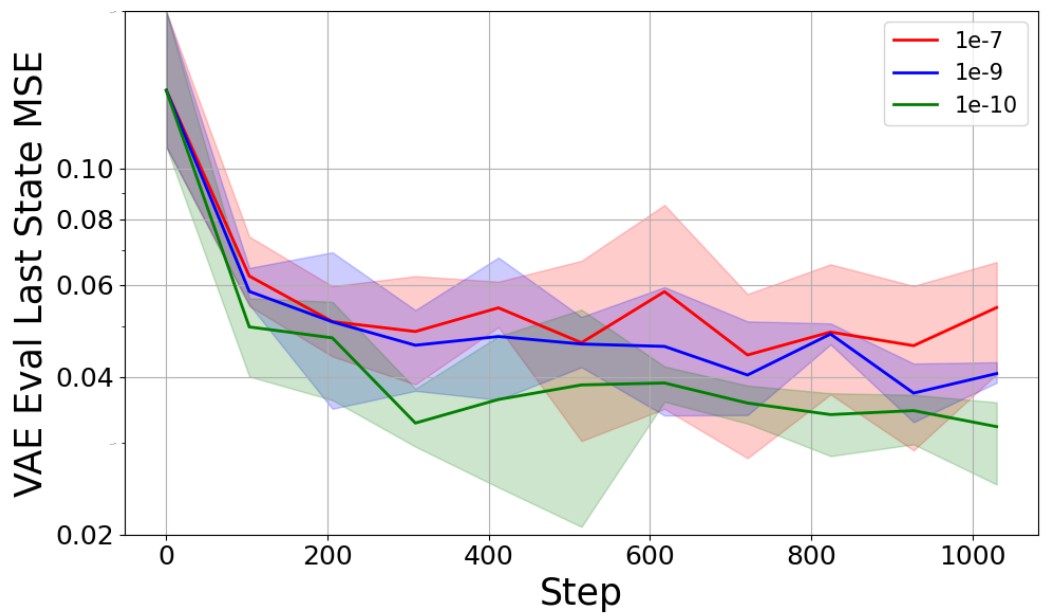

Figure 10: **Effect of KL coefficient**

fig. 10 in fig. 11 shows this metric across 3 seeds during training. Lower $\beta_{\mathrm{KL}}$ values result in lower final-state MSE, indicating better trajectory reconstruction. This is due to a more expressive, multi-modal latent space made possible by weaker regularization, without compromising sampling, as diffusion still operates effectively within this space. Visualizations are provided below in fig. 11. Based on these results, we use $\beta_{\mathrm{KL}} = 1e{-}10$ in all PushT experiments.

Following the KL ablation experiment above, we analyzed the latent space of the encoded trajectories with PCA, similar to that performed in section A.6. The three plots in fig. 11, show that the trajectory encodings get closer and lose behavioral diversity when the KL coefficient is high.

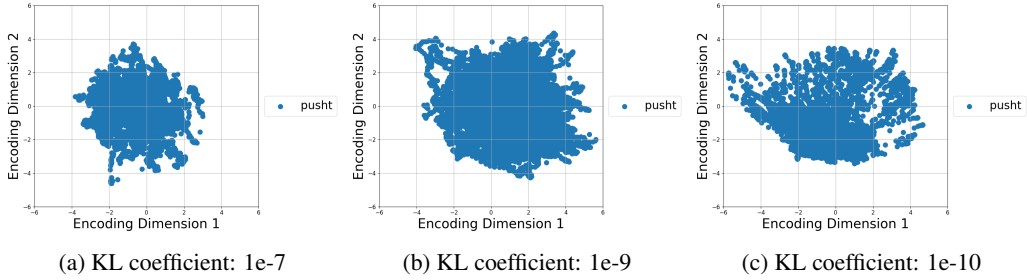

(a) KL coefficient: 1e-7      (b) KL coefficient: 1e-9      (c) KL coefficient: 1e-10

Figure 11: Latent space representation of PushT trajectories at different KL coefficients

## A.5 METAWORLD TASK DESCRIPTIONS

## A.6 EFFECT OF TRAJECTORY SNIPPING ON LATENT REPRESENTATIONS

For most robotics use cases, it is impossible to train on long trajectories due to the computational limitations of working with large batches of long trajectories. In some cases, it may also be beneficial to generate locally optimum policies for shorter action horizons (as done for experiments presented in section 4.1.1). Therefore, we analyze the effect of sampling smaller sections of trajectories from the dataset. After training a VAE for the D4RL half-cheetah dataset on three policies (expert, medium, and random), we encode all the trajectories in the mixed dataset to the latent space. We then perform Principal Component Analysis (PCA) on this set of latents and select the first two principal components. fig. 12a shows us a visualization of this latent space. We see that the VAE has

| Task | Description |
|------|-------------|
| Window Open | Push and open a window. Randomize window positions |
| Door Open | Open a door with a revolving joint. Randomize door positions |
| Drawer Open | Open a drawer. Randomize drawer positions |
| Dial Turn | Rotate a dial 180 degrees. Randomize dial positions |
| Faucet Close | Rotate the faucet clockwise. Randomize faucet positions |
| Button Press | Press a button. Randomize button positions |
| Door Unlock | Unlock the door by rotating the lock clockwise. Randomize door positions |
| Handle Press | Press a handle down. Randomize the handle positions |
| Plate Slide | Slide a plate into a cabinet. Randomize the plate and cabinet positions |
| Reach | Reach a goal position. Randomize the goal positions |

Table 4: Metaworld task descriptions and randomization settings

learned to encode the three sets of trajectories to be well separable. Next, we run the same experiment, but now we sample trajectory snippets of length 100 from the dataset instead of the full-length (1000) trajectories. fig. 12b shows us the PCA on the encoded latents of these trajectory snippets. We see that the separability is now harder in the latent space. Surprisingly, we noticed that after training our VAE on the snippets, the decoded policies from randomly snipped trajectories were still faithfully behaving like their original policies. We believe that this is because the halfcheetah env is a cyclic locomotion task, and all trajectory snippets have enough information to indicate its source policy. More dimensions of the PCA are shown in fig. 13.

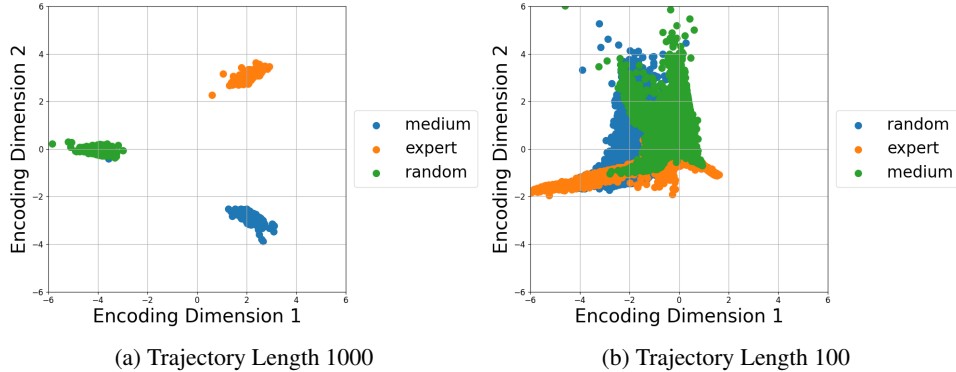

(a) Trajectory Length 1000      (b) Trajectory Length 100

Figure 12: **Effect of trajectory snipping** in HalfCheetah. Top two principal components of the latent.

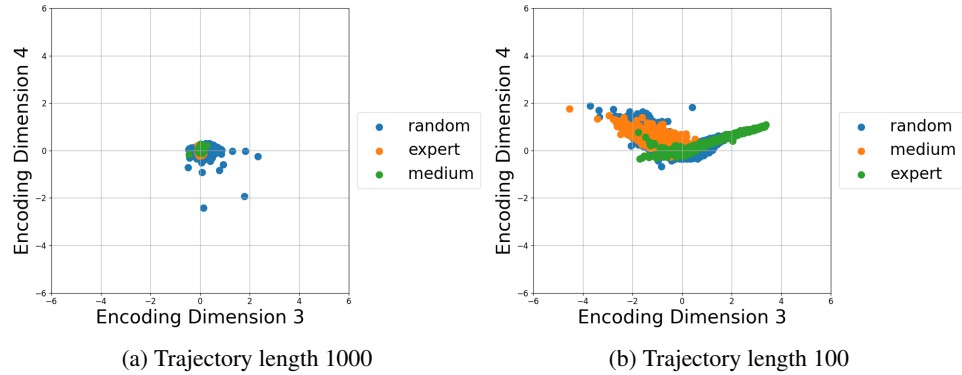

(a) Trajectory length 1000      (b) Trajectory length 100

Figure 13: **Effect of trajectory snipping** in HalfCheetah. Top third and fourth principal components of the latent.

To validate this hypothesis, we analyze our method on trajectory snippets for non-cyclic tasks. We choose the MT10 suite of tasks in Metaworld (Yu et al., 2020) (note that these are different from the original 10 tasks discussed in the rest of the paper. We utilize the hand-crafted expert policy for each of the tasks in MT10 to collect trajectory data. For each task, we collect 1000 trajectories of length 500.

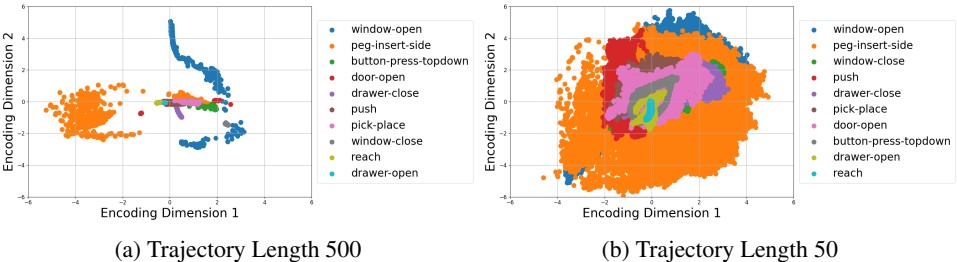

(a) Trajectory Length 500        (b) Trajectory Length 50

Figure 14: **Effect of trajectory snipping** in MT10. Top two principal components of the latent.

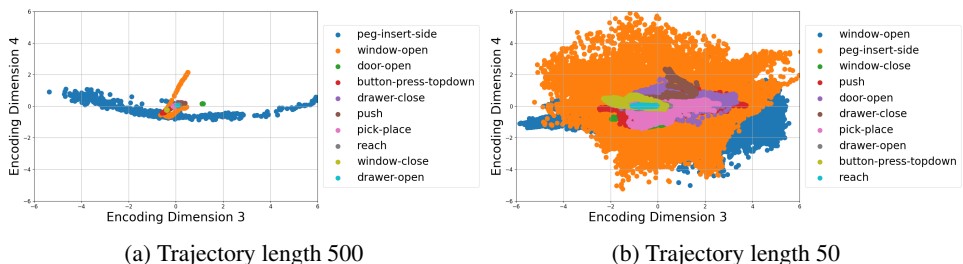

(a) Trajectory length 500        (b) Trajectory length 50

Figure 15: **Effect of trajectory snipping** in MT10. Top third and fourth principal components of the latent.

fig. 14a shows the principal components of the latents of the full trajectories in the dataset, and fig. 14b shows the same for the split trajectories. We can see that the separability of different tasks is much harder in this case. More dimensions of the PCA are shown in fig. 15b. Further, we noticed that the decoded policies from the trajectory snippets did not perform as well as the original policies - for the same decoder size as the half cheetah task. This validates our hypothesis that the snippets are unable to reproduce the original policy for non-cyclic tasks. To have the same degree of behavior reconstruction as the half-cheetah tasks, we need a larger decoder model. This is discussed in section A.4.

## A.7 BEHAVIOR RECONSTRUCTION ANALYSIS

Here, we ask – Does WARPD reconstruct the original policies and reproduce diverse behaviors?

### A.7.1 LOCOMOTION

First, we analyze the behavior reconstruction capability of different components of WARPD in locomotion domains. For this experiment, we use the halfcheetah dataset from D4RL (Fu et al., 2020). The parameters used for this experiment are shown in section A.8.5. Each trajectory in this dataset has a length of 1000. We combine trajectory data from three original behavior policies provided in this dataset: expert, medium, and random. Following (Batra et al., 2023), we track the foot contact timings of each trajectory as a metric for measuring behavior. For each behavior policy, we get 32 trajectories. These timings are normalized to the trajectory length and are shown in fig. 16. For each plot, the x-axis denotes the foot contact percentage of the front foot, while the y-axis denotes the foot contact percentage of the back foot.

We first visualize the foot contact timings of the original policies in fig. 16a. We see that different running behaviors of the half cheetah can be differentiated in this plot. Then, we train the VAE model on this dataset to embed our trajectories into a latent space. We then apply the hypernetwork

decoder to generate policies from these latents. These policies are then executed on the halfcheetah environment, to create trajectories. We plot the foot contact timings of these generated policies in fig. 16b. We see that the VAE captures each of the original policy's foot contact distributions, therefore empirically showing that the assumption $p_{\phi_{dec}}(\theta \mid z) = \delta(\theta - f_{\phi_{dec}}(z))$ is reasonable. Then, we train a latent diffusion model conditioned on a behavior specifier (i.e., one task ID per behavior). In fig. 16c, we show the distribution of foot contact percentages of the policies generated by the behavior specifier conditioned diffusion model. We see that the diffusion model can learn the conditional latent distribution well, and the behavior distribution of the decoded policies of the sampled latent matches the original distribution. Apart from visual inspection, we also track rewards obtained by the generated policies and empirically calculated Jensen Shannon Divergence between the original and obtained foot contact distributions and observe that WARPD maintains behavioral diversity in this locomotion task. See below for more details.

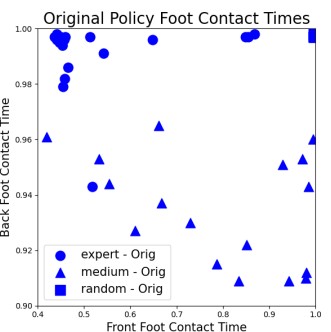 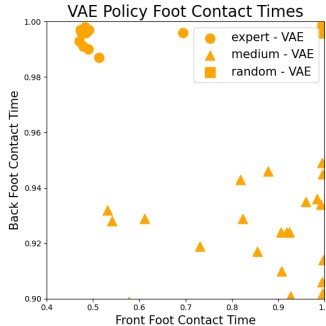 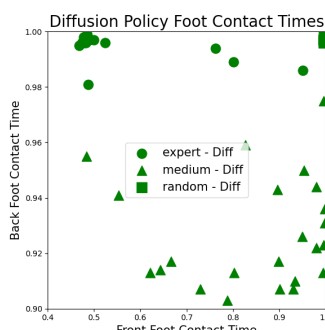

(a) Original policies that provide the trajectory dataset.

(b) VAE generated policies from trajectories.

(c) Diffusion generated policies from trajectories.

Figure 16: **Foot-contact times shown for various trajectories on the Half Cheetah task**. We use foot contact times as the chosen metric to show different behaviors for the half cheetah run task by different policies. The first plot on the left shows the distribution of foot contact percentages for each of the three original policies. The second plot in the center denotes the foot contact percentages for the policies generated by the trained VAE when provided each original policy's entire trajectory. The third plot on the right denotes the foot contact percentages for the policies generated by the diffusion model, trained without any task conditioning.

We can analyze the behavior reconstruction capability of WARPD by comparing the rewards obtained during a rollout. The VAE parameters used for this experiment are shown in section A.8.5. fig. 17 shows us the total objective obtained by the original, VAE-decoded, and diffusion-denoised policies. We see that the VAE-decoded and diffusion-generated policies achieve similar rewards to the original policy for each behavior.

Apart from these plots, we use Jensen-Shannon divergence to quantify the difference between two distributions of foot contact timings. table 5 shows the JS divergence between the empirical distribution of the foot contact timings of the original policies and those generated by WARPD. The lower this value is, the better. As a metric to capture the stochasticity in the policy and environment, we get the JS divergence between two successive sets of trajectories generated by the same original policy, which we shall denote SOS (Same as source). A policy having a JS divergence score lesser than this value indicates that that policy is indistinguishable from the original policy by behavior. As a baseline for this experiment, we train a large (5-layer, 512 neurons each) behavior-conditioned MLP on the same mixed dataset with MSE loss. We see that policies generated by WARPD consistently achieve a lower JS divergence score than the MLP baseline for expert and medium behaviors. The random behavior is difficult to capture as the actions are almost Gaussian noise. Surprisingly, for the HalfCheetah environment, policies generated by WARPD for expert and medium had lower scores than SOS, making it behaviorally indistinguishable from the original policy.

### A.7.2 MANIPULATION

To verify the behavior reconstruction capabilities of WARPD in manipulation, we also experiment on the D4RL Adroit dataset (Rajeswaran et al., 2018). We choose a tool use task, where the agent

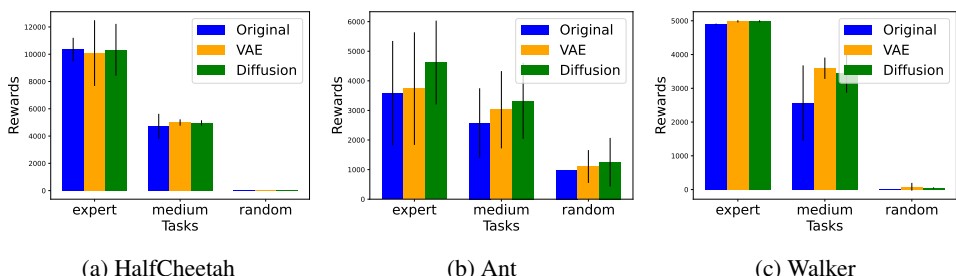

(a) HalfCheetah        (b) Ant        (c) Walker

Figure 17: **Reconstruction Rewards**: For each of the 3 environments shown above, the generated policy from trajectory decoded VAE and task-conditioned diffusion model, achieves similar total objective as the original policies. Each bar indicates the mean total objective obtained with error lines denoting the standard deviation.

| Environment | Source Policy | Target Policy | | |
|---|---|---|---|---|
| | | SOS | MLP | WARPD |
| Ant | Expert | $0.187 \pm 0.142$ | $1.272 \pm 0.911$ | $0.510 \pm 0.159$ |
| | Medium | $0.624 \pm 0.232$ | $1.907 \pm 0.202$ | $1.328 \pm 0.283$ |
| | Random | $1.277 \pm 1.708$ | $4.790 \pm 0.964$ | $8.859 \pm 0.792$ |
| HalfCheetah | Expert | $0.158 \pm 0.146$ | $2.810 \pm 1.139$ | $0.088 \pm 0.050$ |
| | Medium | $0.275 \pm 0.196$ | $0.692 \pm 0.787$ | $0.194 \pm 0.157$ |
| | Random | $0.0467 \pm 0.009$ | $0.11 \pm 0.009$ | $0.104 \pm 0.0187$ |
| Walker2D | Expert | $0.342 \pm 0.329$ | $2.879 \pm 1.493$ | $1.093 \pm 0.310$ |
| | Medium | $0.078 \pm 0.058$ | $0.165 \pm 0.126$ | $0.155 \pm 0.091$ |
| | Random | $0.080 \pm 0.004$ | $60.514 \pm 52.461$ | $2.776 \pm 1.260$ |

Table 5: **Behavior Reconstruction**: JS divergence between foot contact distributions from source and target policies. The lower the value, the better.

must hammer a nail into a board. We utilize their 5000 expert and 5000 human-cloned trajectories, to train our WARPD model. The implementation details are in section A.8.6. Then, we evaluate the behavior of the original and generated policy on the following metrics: **Mean object height** - Average height of the object during eval; **Alignment error (goal distance)** - Mean distance between the target and the final goal position; **Max nail impact** - Maximum value of the nail impact sensor during eval; **Contact ratio** - Fraction of time steps where the nail impact sensor value exceeds 0.8; **Object manipulation score** - Proportion of time steps where the object height exceeds 0.04 meters. From fig. 18, we can see that the policy generated by WARPD behaves similarly to the original policy.

## A.8 IMPLEMENTATION DETAILS

The following are the hyperparameters we use for our experiments:

### A.8.1 BASELINE DIFFUSION POLICY MODEL

To train the diffusion policy baseline model shown in fig. 6, we utilize the training script provided by the authors of DP here:

https://colab.research.google.com/drive/1gxdkgRVfM55zihY9TFLja97cSVZOZq2B?usp=sharing.

To set the model size we use the following parameters:

For the ablation described in section A.3, we use a transformer architecture, the details of which are:

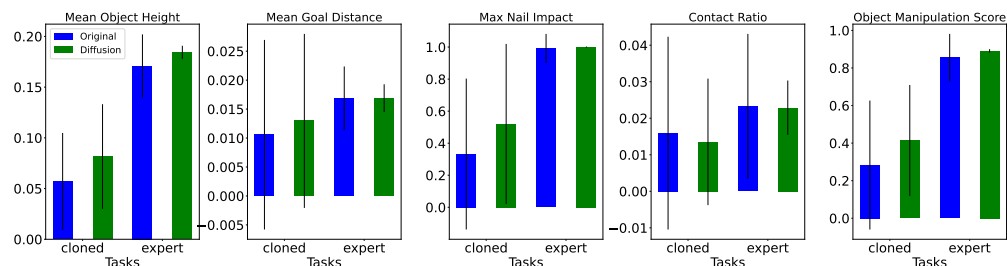

Figure 18: **Behavior Reconstruction for Manipulation**: We track these metrics on the Adroit hammer task, and the WARPD-generated policy behaves similarly to the original policy. The 'cloned' bars represent metrics with respect to a human demonstration behavior cloned policy, and 'expert' bars represent metrics from an RL-trained policy.

| Size | Diffusion Step Embed Dim | Down Dims | Kernel Size |
|------|--------------------------|-----------|-------------|
| extra-small: (s) | 64 | [16, 32, 64] | 5 |
| small: (s) | 256 | [32, 64, 128] | 5 |
| large: (m) | 256 | [128, 256, 256] | 5 |
| large: (l) | 256 | [256, 512, 1024] | 5 |
| extra large: (xl) | 512 | [512, 1024, 2048] | 5 |

Table 6: Architectural configurations for the ConditionUnet1D Diffusion Policy (DP) across different model sizes.

| Size | Diffusion Step Embed Dim | Model Dim | # Layers | # Heads |
|------|--------------------------|-----------|----------|---------|
| extra-small: (xs) | 64 | 64 | 3 | 2 |
| small: (s) | 128 | 128 | 4 | 4 |
| medium: (m) | 256 | 256 | 6 | 8 |
| large: (l) | 256 | 512 | 8 | 8 |
| extra-large: (xl) | 512 | 768 | 12 | 12 |

Table 7: Architectural configurations for Transformer-based Diffusion models across different model sizes.

### A.8.2 VAE ENCODER DETAILS

For the encoder, we first flatten the trajectory to form a one-dimensional array, which is then fed to a Multi-Layer Perceptron with three hidden layers of 512 neurons each.

### A.8.3 VAE HYPERNETWORK DECODER SIZE CHARACTERIZATION

For the hypernetwork, we utilize an HMLP model (a full hypernetwork) from the https://hypnettorch.readthedocs.io/en/latest/ package with default parameters. We condition the HMLP model on the generated latent of dimension 256. To vary the size of the decoder, as explained in section A.4, we set the hyperparameter in the HMLP as shown in table 8

| Size | No. of parameters | layers |
|------|-------------------|--------|
| xs | 3.9M | [50, 50] |
| s | 7.8M | [100, 100] |
| m | 15.6 M | [200, 200] |
| l | 31.2M | [400, 400] |

Table 8: VAE size varying parameters

### A.8.4 DIFFUSION MODEL PARAMETERS

For all our experiments, we utilize the same ConditionalUnet1D network from Chi et al. (2024) as the diffusion model. This is the same as the DP-medium (m) model described in section A.8.1.

### A.8.5 MUJOCO LOCOMOTION TASKS

We use the following hyperparameters to train VAEs for all D4RL mujoco tasks shown in the paper. To show the effect of shorter trajectories in section A.6, we change the Trajectory Length to 100.

| Parameter | Value |
|---|---|
| Trajectory Length | 1000 |
| Batch Size | 32 |
| VAE Num Epochs | 150 |
| VAE Latent Dimension | 256 |
| VAE Decoder Size | s |
| Evaluation MLP Layers | {256, 256} |
| VAE Learning Rate | $3 \times 10^{-4}$ |
| KL Coefficient | $1 \times 10^{-6}$ |
| Diffusion Num Epochs | 200 |

Table 9: Mujoco locomotion hyperparameters.

### A.8.6 ADROIT HAMMER TASK

We use the same hyperparameters as table 9 and override the following hyperparameters to train VAEs for the D4RL Adroit hammer task shown in the paper.

| Parameter | Value |
|---|---|
| Trajectory Length | 128 |
| VAE Num Epochs | 20 |
| Diffusion Num Epochs | 10 |

Table 10: Adroit hammer hyperparameters.

Further, for the experiment where we show the hammer task can be composed of sub-tasks, we change the Trajectory Length to 32 to enable WARPD to learn the distribution of shorter horizon policies.

### A.8.7 PUSHT AND ROBOMIMIC WARPD

For all the experiments shown in section 4.1.1, we use the same hyper-parameters described in table 9, and override the following:

| Parameter | Value |
|---|---|
| Trajectory Length | 256 |
| VAE Num Epochs | 1000 |
| Diffusion Num Epochs | 1000 |
| Diffusion Model size | l |
| VAE Decoder Size | l |
| VAE KL coefficient | $1e - 10$ |

Table 11: PushT WARPD hyperparameters.

### A.8.8 METAWORLD TASKS

For all the experiments shown in section 4.1.2, we use the same hyper-parameters described in table 9, and override the following:

| Parameter | Value |
|---|---|
| Trajectory Length | 500 |
| VAE Num Epochs | 100 |
| Diffusion Num Epochs | 100 |
| VAE Decoder Size | `xs` |

Table 12: Metaworld hyperparameters.

To show the effect of shorter trajectories in section A.6, we change the Trajectory Length to 50.

## A.9 COMPUTE RESOURCES

Each VAE and diffusion experiment was run on jobs that were allocated 6 cores of a Intel(R) Xeon(R) Gold 6154 3.00GHz CPU, an NVIDIA GeForce RTX 2080 Ti GPU, and 108 GB of RAM.

Our observations indicate that the training time for each component of WARPD is approximately equivalent to that of a full DP training run: $traintime(\text{DP}) \simeq traintime(\text{VAE}_{WARPD}) \simeq traintime(\text{Diffusion}_{WARPD})$

Therefore, the total training time for WARPD is approximately $2 * traintime(DP)$. To provide a concrete example, for the PushT task with image observations, using a compute configuration of a Tesla P100-PCIE-16GB GPU, 16 Intel Xeon Gold 6130 CPU cores, and 64GB RAM, we observed the following wall-clock training times:

- 2000 epochs of DP training: 13 hours 8 minutes
- 1000 epochs of WARPD's VAE training: 12 hours 32 minutes
- 1000 epochs of WARPD's diffusion training: 13 hours 37 minutes

