# OpenReview forum: "WARPD – World-model Assisted Reactive Policy Diffusion"
_ICLR.cc/2026/Conference — Submitted to ICLR 2026_

### Official Review · Reviewer_qqrx · 2025-10-29

**Soundness:** 3
**Presentation:** 3
**Contribution:** 3
**Rating:** 6
**Confidence:** 3

**Summary:**

This paper introduce world-model assisted reactive policy diffusion, which learns a closed-loop controller rather than a trajectory predictor. Apart from learning a policy parameter encoder and decoder with VAE, this paper’s major contribution is introduce a co-trained world model to provide auxiliary loss to ensure policy consistency and robustness.

**Strengths:**

- the proposed warpd demonstrate significant speed up for diffusion model, highlight the efficiency of reactive policy.

- the proposed method also preserve the diversity of expert demonstrations, indicating the importance of VAE encoder.

- the modeified ELBO is nealty and the decomposed loss function is well motivated.

**Weaknesses:**

- since world model is merely trained on data collected with expert, it might be unreliable for OOD states, which is precisely where stability is needed. This paper doesn’t clearly show how the world model improve robustness under such conditions. Quantitive analysis of modelling error v.s. policy performance would help.

- WARDP encoder trajectories into latent rather than policy weights used in standard hypernetwork. Clarifying what the latent represents and comparing to baselines like [1] would make contributino cleaner.


[1] Reactive Diffusion Policy: Slow-Fast Visual-Tactile Policy Learning for Contact-Rich Manipulation. Xue et al.

**Questions:**

- can the authors evaluate WARPD on more challenging tasks (e.g., Robomimic Tool Hang, Transport) or with varied expert quality and input modalities? The current tasks (Can, Lift) are relatively easy and don’t fully test robustness since both of them are relative easy to solve even with regression.

- PushT and Robomimic rely on high-frequency position control. Can the authors show that WARPD still outperforms diffusion policies when this stabilizing layer is removed or control frequency is reduced? (i.e. directly do torque control, where the station becomes unstable)

---

> ### Author Response · Authors · 2025-12-02
>
> # W: World model trained only on expert data; unclear OOD robustness
>
> - We agree a world model trained only on expert trajectories can be inaccurate far from the data manifold. Our design explicitly does **not** rely solely on the world model in such regions: the behavior cloning loss always supervises the generated policy on true expert actions, so **in the worst case WARPD behaves like a diffusion-augmented BC method rather than a purely model-based controller. This matches prior work showing model-based IL can reduce covariate shift in imitation learning [2,3].**
> - **The ablation WARPD vs WARPD w/o WM already gives partial robustness evidence: on Robomimic, where covariate shift is stronger, full WARPD significantly outperforms WARPD w/o WM, while on PushT (milder shift) the gap is smaller.** This is consistent with the world model primarily helping when off-dataset dynamics matter.
> - A full quantitative mapping from modeling error to performance is indeed valuable but orthogonal to our main question (policy-space vs trajectory-space diffusion). We treat the existing ablation as initial evidence that the world model improves robustness in practice, while acknowledging that a deeper analysis is interesting future work outside the current scope.
>
> ---
>
> # W: Latent representation vs standard hypernetworks; relation to RDP [1]
>
> - In WARPD, the encoder maps trajectories → latent \(z\), and a hypernetwork maps \(z \rightarrow \theta\) (policy parameters). Thus \(z\) is a **behavior embedding**: it captures task- and style-level properties of expert trajectories that are then instantiated as a specific closed-loop policy via the hypernetwork.
> - This differs from “standard” hypernetworks that take a task/context embedding and directly output \(\theta\) without explicitly modeling a distribution over behaviors. WARPD models a distribution over \(z\) via diffusion, enabling multimodal behavior families rather than a single point estimate per context.
> - We thank the reviewer for pointing out RDP [1] and have cited it (see Sections 2.1, 3.3, and Table 1). **Philosophically, RDP and WARPD share a slow–fast hierarchy: a slower process performs diffusion, and a faster, smaller policy performs closed-loop control. Compared to RDP, which uses diffusion to generate a latent representation of **action chunks**, WARPD uses diffusion to generate a latent representation of **policy parameters**. The LDP baseline in our experiments, which diffuses latent action representations, shows that this strategy is suboptimal relative to policy-space diffusion.**
>
> ---
>
> # Q: Harder tasks, varied experts/modalities
>
> - **We chose Lift and Can because they are widely used, standardized Robomimic benchmarks with well-understood difficulty and baselines**, enabling direct comparison and careful study of horizon and perturbation effects. They also balance task complexity with the ability to perform extensive sweeps (horizons, perturbations, compute).
> - We view WARPD as a general framework that can be applied to harder tasks, varied expert quality, and additional modalities (e.g., extending the PushT image setup). **Systematically exploring all these axes would substantially expand the empirical scope and is beyond this initial study**, which is primarily aimed at contrasting policy-space vs trajectory-space diffusion under controlled conditions.
>
> ---
>
> # Q: Effect of low-level stabilizing control / control frequency
>
> - Our experiments adopt the standard control abstractions of the underlying benchmarks (position control for PushT and Robomimic), so comparisons to existing work and baselines (DP, LDP) remain meaningful. In this setting, the low-level controller is part of the environment dynamics shared by all methods.
> - The central comparison is *where* diffusion operates (policy vs trajectories) under matched environments and control interfaces. In all reported settings, WARPD and the diffusion baselines share the same low-level stabilizing layer, so the observed robustness and compute improvements stem from the generative representation rather than actuation differences.
> - **Studying direct torque control or much lower control frequencies would introduce an additional axis of difficulty and engineering (joint-level stability, safety constraints) that is orthogonal to our main questions.** We consider this an interesting, but separate, setting from the standardized high-frequency benchmarks used here.
>
> ---
>
> # REFS:
> [1] Xue et al., *Reactive Diffusion Policy: Slow-Fast Visual-Tactile Policy Learning for Contact-Rich Manipulation.*
> [2] Hu, Anthony, et al. "Model-based imitation learning for urban driving." *Advances in Neural Information Processing Systems* 35 (2022): 20703–20716.
> [3] Popov, Alexander, et al. "Mitigating covariate shift in imitation learning for autonomous vehicles using latent space generative world models." arXiv preprint arXiv:2409.16663 (2024).

---

### Official Review · Reviewer_5bWh · 2025-10-30

**Soundness:** 2
**Presentation:** 2
**Contribution:** 1
**Rating:** 2
**Confidence:** 4

**Summary:**

This paper focuses on imitation learning for robotics—specifically in manipulation and locomotion—using diffusion models. It notes that while diffusion policies effectively capture multimodal action distributions and support generalized policy behavior, they suffer from large model sizes and slow inference speeds, which are critical limitations in tasks requiring high control frequencies. To address these issues, the authors propose WARPD (World Model Assisted Reactive Policy Diffusion), a method that integrates a learned world model to assist a diffusion-based policy, enabling fast and reactive control while retaining the advantages of diffusion-based action generation. The work seeks to bridge diffusion-based imitation learning with model-based prediction to achieve more responsive policies. It clearly identifies inference latency and model size as key bottlenecks of diffusion policies in high-frequency robotic control settings and underscores the limitations of trajectory-generation Diffusion Policy approaches in such scenarios.

**Strengths:**

The paper demonstrates significant originality by reframing diffusion-based imitation learning to generate closed-loop policies directly in parameter space, a conceptual shift that effectively preserves multimodality while overcoming the latency and trajectory-tracking pitfalls of prior methods. It astutely diagnoses the underexplored but critical trade-off between action horizon and tracking accuracy in trajectory diffusion under high-frequency constraints. The integration of a learned world model with latent diffusion to enable reactive control is a creative synthesis that cohesively addresses inference speed and stability.

**Weaknesses:**

1. **Novelty and positioning need sharper evidence**
The core idea—use latent diffusion plus a world model to generate closed‑loop policies directly in parameter space, bypassing trajectory generation—is interesting but overlaps conceptually with prior lines on context/hypernetwork-conditioned policies and model-assisted imitation learning. The paper should more clearly articulate what is fundamentally new versus: (a) Diffusion Policy variants that reduce query counts via action chunking or one step diffusion methods; (b) hypernetwork/meta-learning approaches that generate policy parameters from context; (c) model-based IL/RL with world-model regularization. Concretely: add head-to-head comparisons and an explicit “differences vs. closest work” table. Include works on policy parameter generation via hypernetworks/meta-learning and diffusion-based control to delineate WARPD’s distinct contribution. When claiming to “bypass trajectory generation,” clarify how this differs in practice from generating shorter horizons plus feedback. Provide a targeted ablation replacing diffusion with simpler generators (e.g., VAE or normalizing flows) for the same parameter-space policy generation to show diffusion is necessary for the claimed benefits, not just any latent generator.
2. **Real-world deployment readiness not yet demonstrated**
If all results are in simulation, add at least one real-robot evaluation or a sim-to-real transfer test, reporting latency, success, and safety incidents. Even a short, controlled study would substantiate the practical relevance of the claimed speed and reactivity.

**Questions:**

The main issues and suggestions have been outlined in the Weakness section. Additional points include:

The experimental section should be enhanced by adding comparisons with other state-of-the-art Diffusion Policy and Model-based RL methods.

Please use vector-based figures for all illustrations to ensure optimal clarity and scalability.

---

> ### Author Response · Authors · 2025-12-02
>
> # W: Novelty and positioning
>
> - **Positioning vs Diffusion Policy.**
>   WARPD directly targets the trajectory–vs–policy diffusion trade-off: instead of diffusing action chunks, we diffuse in a latent *policy* space and decode full closed-loop policies that run for long horizons at high control frequency. In contrast, DP variants reduce query count via shorter horizons or one-step/noise-conditioned schemes but still generate open-loop trajectories between diffusion calls. We compare head-to-head with DP and LDP using shared diffusion backbones and matched hyperparameters to isolate policy-space vs trajectory-space diffusion.
>
> - **Positioning vs hypernetworks/meta-learning.**
>   Prior hypernetwork/meta-learning work generates policy parameters from explicit policy datasets or task/task-context embeddings (e.g., Hegde et al., 2023; von Oswald et al., 2020). WARPD instead:
>   - encodes *demonstration trajectories*, not policies;
>   - learns a latent policy distribution via diffusion; and
>   - optimizes generated policies with a world-model-augmented imitation objective (BC + rollout).
>   The novelty is trajectory-to-policy *generation* via a latent diffusion prior, rather than parameter reconstruction or few-shot adaptation from existing policies.
>
> - **Positioning vs model-based IL/RL.**
>   Model-based IL/RL (e.g., Dreamer-style) uses world models to optimize a single policy per task, typically with online RL. WARPD uses the world model differently: to regularize a *distribution* over policies learned from offline demonstrations and to correct generated policies via the rollout loss L_RO, integrated into the mELBO-inspired Stage-1 objective that ties the encoder, policy decoder, and dynamics.
>
> - **On “bypassing trajectory generation.”**
>   By this, we mean the diffusion model never outputs action sequences; it outputs latents that decode to policy parameters. The resulting policies compute actions reactively at every time step. Even with short horizons, DP still executes a fixed action chunk between diffusion calls, whereas WARPD evaluates a state-feedback policy at every step.
>
> - **Diffusion vs simpler generators.**
>   Our empirical study stays within the diffusion-based IL family: DP and LDP vs WARPD, plus an MLP baseline showing that removing multimodal generative modeling hurts performance on tasks like PushT. We do not claim diffusion is uniquely necessary among all latent generators (e.g., flows, VAEs); rather, we show that *within* diffusion, modeling in policy space brings robustness and efficiency gains over trajectory-space diffusion. Exploring alternative generators for policy-space modeling is orthogonal and left for future work.
>
> - **Core contribution clarification.**
>   A central contribution is neural *parameter* generation from trajectory datasets in a purely imitation-learning setting. To our knowledge, this has not been done before; we lay the groundwork via the theoretical derivation in the paper. **To further clarify this, we have added Section 3.3 and Table 1 to the revised manuscript.**
>
> # W: Real-world deployment readiness not yet demonstrated
>
> - We agree results are simulation-only and explicitly state this in the Limitations section. PushT, Robomimic, and MetaWorld enable controlled evaluation of long horizons, perturbations, and FLOPs/step that would be hard to sweep systematically on hardware. Real deployment additionally requires robot-specific perception stacks, safety constraints, and reset mechanisms.
> - We treat our simulation results as a step toward real-robot deployment, not as a claim of immediate readiness. **The practically relevant aspects, robustness under perturbations and low per-step inference cost, are measured directly** (e.g., improved long-horizon robustness on PushT/Robomimic and ~45× lower FLOPs/step than DP on MetaWorld at comparable success).
>
> ---
>
> # Q: More comparisons with SOTA Diffusion Policy and model-based RL; vector-based figures
>
> - Our experimental focus is diffusion-based imitation and the contrast between:
>   - trajectory-space diffusion (DP, LDP), and
>   - policy-space diffusion (WARPD).
>   We therefore use DP and LDP as state-of-the-art diffusion baselines with matched architectures/hyperparameters, plus MLP policies as non-diffusion baselines, to isolate the effect of moving diffusion from trajectory space to policy-parameter space.
> - Model-based RL methods such as DreamerV3 or UniZero operate in a different setting (online RL, reward-driven optimization, task-specific data collection). We discuss them in Related Work on world models and hypernetworks, but they are not directly comparable IL baselines in our offline, demonstration-driven setting.
> - We will adopt the suggestion on figures as a presentational improvement: all plots will be rendered in vector-based formats in the final version for better clarity and scalability.

---

### Official Review · Reviewer_qYVo · 2025-11-01

**Soundness:** 3
**Presentation:** 3
**Contribution:** 2
**Rating:** 4
**Confidence:** 3

**Summary:**

This paper proposes WARPD, a diffusion-based imitation learning framework that generates closed-loop policies (network weights) rather than open-loop action trajectories. Instead of predicting a sequence of actions like Diffusion Policy (DP), WARPD samples policy parameters via latent diffusion and decodes them using a hypernetwork conditioned on states or task identifiers. A learned world model is used during training to align policies with realistic system dynamics and correct drift from the dataset distribution.

WARPD aims to address three main limitations in trajectory diffusion approaches:

(1) Slow inference from repeatedly denoising long action sequences,

(2) Tracking errors from large action horizons, and

(3) Poor robustness under perturbations due to open-loop control.
The authors show experiments on PushT, Robomimic (Lift & Can), and 10 MetaWorld tasks, reporting better long-horizon robustness, ~45× lower inference FLOPs, and competitive success rates.

**Strengths:**

Novel shift from action-space to parameter-space diffusion
Instead of diffusing trajectories, WARPD diffuses policy weights via a hypernetwork, which naturally enables closed-loop control and removes the need to regenerate trajectories at every timestep.

Addresses the core weakness of diffusion policies (latency & drift)
By producing a policy that runs reactively at high frequency, WARPD avoids action-horizon drift and reduces the number of diffusion queries required per episode.

Integration of world model is meaningful (not just architectural trick)
The world model guides the policy to stay in-distribution and helps correct state deviations during training. This makes the approach more grounded than purely behavior cloning + diffusion.

Strong empirical evaluation (robustness + efficiency)

Handles long-horizon perturbed control better than Diffusion Policy (Fig. 3 & Fig. 5)

On MetaWorld multitask settings, achieves ~45× lower inference FLOPs at comparable success rates (Fig. 6)

Demonstrates both state-based and vision-based versions, which increases credibility of generalization.

Clear theoretical formulation (modified ELBO + hypernetwork objective)
The paper derives a structured learning objective combining behavior cloning, reconstruction via world model, and latent diffusion. This helps show it’s not just a heuristic but grounded in generative modeling.

Behavior diversity captured in latent space
The t-SNE result showing skills clustered by human demonstrators (Fig. 7) suggests WARPD can learn structured behavioral variability without explicit supervision.

**Weaknesses:**

Still relies on low-dimensional state observations in main experiments
Vision-based results only appear in a small section (Table 1), using frozen encoders rather than full end-to-end visual policy generation.

World model quality is critical but not deeply analyzed
If the learned dynamics are inaccurate, does WARPD degrade? The paper does not provide failure cases or show robustness to model inaccuracies.

Training complexity and compute overhead are downplayed
Although inference is cheaper, training requires three components (VAE + world model + diffusion). No wall-clock training cost or GPU hours are reported.

Policy generation via hypernetworks may limit expressiveness
Only MLP policies are considered. It's unclear whether this method scales to transformers or visuomotor architectures like RT-1, Octo, or Diffusion Policy with image inputs.

Comparison baselines could be broader
Missing comparisons with model-based policy generation approaches like DreamerV3, HyperPPO, UniZero, or transformer-based policy distillation.

Limited real-robot or real-world experiments
All experiments are simulated environments (PushT, MetaWorld, Robomimic). No deployment on physical robots, making practicality uncertain.

**Questions:**

see weaknesses

---

> ### Author Response · Authors · 2025-12-02
>
> # W: Results on low-dimensional states; limited vision
> - **Our goal is to study policy-space diffusion in a controlled, low-dimensional setting, where robustness and compute efficiency can be measured cleanly.** For this reason, we use state observations in well established benchmarks (PushT state, Robomimic state, MetaWorld).
> - **Section 4.3 presents a proof-of-concept vision setup on PushT images**: we pre-train a vision encoder, feed its embeddings into WARPD, and compare against DP on the same embeddings. WARPD remains more robust under perturbations at horizon 64, suggesting the method transfers to visual embeddings.
> - **WARPD is agnostic to how state embeddings are obtained**: the diffusion model and hypernetwork operate on whatever latent state is provided. End-to-end visuomotor training is compatible in principle but beyond the scope of this first study and left for future work.
>
> # W: World model quality not deeply analyzed
> - **The world model is explicitly ablated: WARPD vs WARPD w/o WM (no rollout loss, L_RO).** On PushT the gap is modest; on Robomimic the full method significantly outperforms WARPD w/o WM. This is consistent with our claim that the world model helps most under stronger covariate shift.
> - Training is structured so WARPD does not collapse when the world model is imperfect:
>   - The world model is warm-started via teacher forcing before L_RO is activated.
>   - The behavior cloning loss always provides a supervised signal from demonstration actions, so in the worst case WARPD behaves like a diffusion-augmented BC model without relying heavily on rollouts.
> - A systematic sweep over artificially degraded world models is an interesting direction for general model-based imitation learning. Still, **current experiments already show that performance degrades gracefully when removing world-model guidance and that the method remains competitive without it.**
>
> # W: Training complexity and compute overhead
> - We discuss additional training overhead in the Limitations section and detail compute resources in Section A.9.
> - **WARPD is aimed at offline imitation settings where (with modern GPUs) training cost is not as important as inference-time compute/latency.** In this regime, the trade-off is favorable: WARPD matches DP on 10 MetaWorld tasks while using ~45× fewer FLOPs per control step, even counting policy generation.
> - The revised text clarifies this trade-off by separating training complexity (comparable to other world-model IL methods) from runtime efficiency, which is the main contribution (see Section 5).
>
> # W: Hypernetwork policies limited to MLPs
> - The hypernetwork design is architecture-agnostic in principle: it maps a fixed latent \(z\) to a vector of parameters.
> - Considering that we already use a hypernetwork module that is already capable of more complicated models, we already note extension to transformer/ViT policies as a natural direction (e.g., chunked/structured hypernetworks, as in von Oswald et al. (2020)).
> - **The PushT Image experiment already couples WARPD with a ResNet18 vision encoder + MLP controller, showing that WARPD can sit downstream of a nontrivial perception stack without changing the core algorithm.**
>
> # W: Baselines could be broader (DreamerV3, HyperPPO,...)
> - **WARPD is not an RL method, but a imitation learning algorithm.**
> - Our focus is imitation learning with diffusion, specifically comparing:
>   - Trajectory-space diffusion (DP, LDP), vs
>   - Policy-space diffusion (WARPD).
>   Baselines are chosen to match this scope: DP and LDP as strong action/latent-action diffusion policies, and an MLP BC baseline.
> - **Methods such as DreamerV3, UniZero, HyperPPO are RL-centric with different training signals (environment interaction with reward data). This is orthogonal to our central question of “trajectory vs. policy diffusion for IL.”**
> - Conceptually, we position WARPD beside these methods in Related Work, emphasizing that our contribution is to combine latent diffusion, world models, and hypernetworks specifically for offline imitation from trajectories.
>
> # W: Limited real-robot / real-world experiments
> - We agree this is a limitation and state it explicitly: all experiments use **established simulated benchmarks** (PushT, MetaWorld, Robomimic) to control variability and enable extensive sweeps over horizons and perturbations.
> - WARPD targets two bottlenecks that matter for real hardware, robustness under perturbations and low per-step inference cost, which are demonstrated in simulation (e.g., robustness on PushT/Robomimic and ~45× lower FLOPs/step on MetaWorld).
> - **Deploying WARPD on physical robots will require integration with robot-specific perception stacks, safety measures, and data-collection procedures. We view the simulation results as an important step toward that goal rather than a claim of solving real-robot deployment.**

---

### Official Review · Reviewer_SEqh · 2025-11-03

**Soundness:** 2
**Presentation:** 3
**Contribution:** 2
**Rating:** 4
**Confidence:** 2

**Summary:**

WARPD proposes a diffusion-based world-model-assisted controller that diffuses latent variables used to generate policy weights via a hypernetwork, rather than directly diffusing actions. The approach is claimed to reduce inference compute while maintaining or improving performance across multi-task manipulation and perturbation settings. A modified ELBO objective combines imitation, world-model rollout, and latent regularization terms.

**Strengths:**

- The paper tackles an underexplored bottleneck in diffusion-based control and provides a plausible path toward efficiency via policy-space generation.

- WARPD combines ideas from world models, latent diffusion, and hypernetwork-based policy generation, forming a coherent if complex system that extends recent model-based imitation learning work.

- The experimental coverage is broad, with consistent reporting of FLOPs, ablations, and hyperparameters.

- The reported performance gains in long-horizon and perturbed settings are practically meaningful.

- The work addresses a timely and relevant problem in large-scale robot policy learning.

**Weaknesses:**

- The system chains together a world model, VAE encoder, diffusion model, and hypernetwork, with partial joint optimization. This makes it difficult to isolate where improvements originate. The theoretical story suggests unified probabilistic modeling, but the actual implementation is a series of independent training stages.

- The paper’s modified ELBO derivation is central to its conceptual framing. However, the KL term is effectively disabled (β ~ 1e-10), meaning the latent space is unregularized. The diffusion model thus learns to imitate arbitrary encoder codes, not a well-defined prior. Sampling from the prior (as the ELBO suggests) would not produce coherent behaviors. Therefore, the method is not truly a probabilistic generative model, and the ELBO justification is more rhetorical than operative. The paper’s central theoretical claims (generative nature, latent consistency, principled regularization) would collapse under this setting.

- I’m not convinced the central architectural choice of diffusing policy weights via a hypernetwork instead of diffusing actions is necessary or clearly beneficial. The paper provides no ablation or evidence that this design improves over simpler schemes (e.g., latent trajectory diffusion). In fact, several implementation choices (KL≈0, world-model rollout, multi-stage training) seem to exist primarily to make this complicated setup trainable. As a result, the method feels overengineered to support this representation.

**Questions:**

1) Given β≈0, are you truly sampling from a generative prior or simply predicting deterministic latents? How does this affect the validity of the modified ELBO derivation?
2) Is the hypernetwork really necessary? Have you compared against diffusing actions or latent trajectories directly, and can you clarify what specific benefit weight diffusion brings?

---

> ### Author Response · Authors · 2025-12-02
>
> # W 1: Many components, staged training, “not truly unified”
> - During Stage 1, the encoder, hypernetwork decoder, and world model are jointly trained with a single loss (behavior cloning + rollout + teacher forcing + KL) on a shared latent representation, so these modules are not independent.
> - The two-stage structure (Stage-1 VAE + world model, Stage-2 prior diffusion) follows standard latent-diffusion practice: first learn a latent space and decoder, then a prior over that space. **This is standard in image, audio, video, and text latent diffusion [1–4].**
> - **Table 2 in Section 4.2 now explicitly maps each component to its ablation/baseline.** Thus, attribution is supported by ablations and baselines (WARPD, WARPD w/o world model, DP, LDP, MLP)
>
> # W 2: β≈0, unregularized latents, ELBO “rhetorical”
> - β is small but not zero and acts as a weak smoothness regularizer rather than enforcing q(z|τ) ≈ N(0, I). Larger β hurts reconstruction and performance (see our β ablation), so we intentionally keep it small while avoiding completely unconstrained latents, a weak-KL regime common in latent diffusion [1–4].
> - The operative prior at generation time is the diffusion model trained on q(z|τ); the Gaussian prior in the modified ELBO is a reference prior for Stage-1 training, not the test-time prior. The modified ELBO is an “ELBO-inspired” objective for the Stage-1 VAE + world model. **After replacing the simple Gaussian with a learned diffusion prior, the closed-form bound changes, but the overall system remains probabilistic.**
> - Empirically, the latent space exhibits meaningful structure (e.g., clustering by operator/behavior quality), showing that even with weak KL, z encodes consistent behavioral semantics.
> - **Figure 16 further shows that WARPD maintains the stochastic behavior distribution.**
>
> # W 3: Necessity of weight diffusion + hypernetwork; “overengineered”
> - We directly compare to action-trajectory and latent-trajectory diffusion: DP (action sequences), LDP (same diffusion backbone as WARPD but decoding action chunks), WARPD w/o world model, and an MLP policy with the same architecture as the generated policies.
> - Across long horizons and perturbations, WARPD consistently outperforms both DP and LDP, despite LDP using the same latent-diffusion machinery, supporting the benefit of generating closed-loop policies rather than trajectories.
> - Weight diffusion via a hypernetwork provides additional efficiency and sharing: diffusion is invoked infrequently to generate policies, while per-step control uses a small MLP, and a single hypernetwork generates many task-specific policies.
> - **The WARPD / WARPD w/o WM / DP / LDP ablations in Table 2 are designed to expose this complexity–benefit tradeoff.**
> - **These components are also motivated by the derivation in Section 3.1.**
>
> # Q 1: β≈0, generative prior, ELBO validity
> - The overall system is stochastic and generative: the diffusion model defines a distribution over latents z; at test time we sample from this diffusion prior, decode to policy weights, and act in the environment.
> - β only affects how the latent space is learned in Stage 1; the test-time prior is the learned diffusion prior, not N(0, I).
> - **The Gaussian prior in the modified ELBO is a training construct, and the modified ELBO should be read as a principled Stage-1 objective rather than a closed-form bound for the full diffusion-augmented model.**
>
> # Q 2: Is the hypernetwork necessary? Benefits of weight diffusion
> - **We compare to DP (actions), LDP (latent trajectories), and MLP policies; LDP is a “latent trajectory diffusion” baseline that mainly differs in decoding actions rather than policy weights. WARPD achieves the best performance in long-horizon, perturbed, and multi-task regimes, where closed-loop policies are particularly advantageous.**
> - The hypernetwork is useful because it (i) enables parameter sharing across tasks and behaviors in weight space, (ii) keeps diffusion in a fixed-dimensional latent space independent of the policy architecture, and (iii) allows expensive diffusion to run infrequently while per-step control uses a small, efficient policy network.
> - These points are summarized in Table 2.
>
> # REFS:
> [1] Rombach, Robin, et al. "High-resolution image synthesis with latent diffusion models." Proceedings of the IEEE/CVF conference on computer vision and pattern recognition. 2022.
> [2] Liu, H., Chen, Z., Yuan, Y., Mei, X., Liu, X., Mandic, D., Wang, W., & Plumbley, M. (2023). AudioLDM: Text-to-Audio Generation with Latent Diffusion Models. Proceedings of the International Conference on Machine Learning, 21450-21474.
> [3] Blattmann, Andreas, et al. "Align your latents: High-resolution video synthesis with latent diffusion models." Proceedings of the IEEE/CVF conference on computer vision and pattern recognition. 2023.
> [4] Lovelace, Justin, et al. "Latent diffusion for language generation." Advances in Neural Information Processing Systems 36 (2023): 56998-57025.

---

### Meta-Review · Area_Chair_Vc35 · 2026-01-07

**Summary:**

WARPD is a diffusion-based imitation learning framework that generates closed-loop policies by diffusing in policy parameter space via a hypernetwork decoder and a learned world model, aiming to improve robustness and reduce inference cost relative to trajectory diffusion. Reviewers generally agreed that the idea of policy-parameter diffusion is conceptually novel, but raised concerns regarding the clarity and necessity of the probabilistic formulation, the complexity and interaction of multiple components, the absence of real-robot validation, significance and clarity of actual contribution. Despite additional explanations and ablations during discussion, major concerns were not fully resolved; therefore recommend rejection at this time.

NOTE: possibly hallucinated references: Hao Zhang, Zhihan Xu, Jian Liu, and Qingzhao Wang. Generalized and efficient planning with scalable latent world models. arXiv preprint arXiv:2406.10667, 2024.

**Reviewer Concerns:**

Concerns largely addressed by the rebuttal
- [Seqh] Many components & staged training; Necessity of weight diffusion and hypernetwork
- [qYVo] Low-dimensional state focus; Role and necessity of the world model; Training complexity & inference efficiency trade-off; Baseline scope  (RL/IL)
- [5bWh] positioning relative to prior work
- [qqrx] trained on expert data / unclear OOD robustness; relation to standard hypernetworks and RDP; Preservation of multimodal behavior

Concerns still outstanding
- [Seqh] Overengineering (partially); Validity of the probabilistic framing; theoretical claims
- [qYVo] Limited analysis of failure modes; Expressiveness of hypernetwork-generated policies; Lack of real-world validation
- [5bWh] novelty claim; lack of real-world validation; contribution significance
- [qqrx] quantitative analysis; Task difficulty and evaluation expansion; Dependence on high-frequency stabilizing control

**Reviewer Scores:**

- [Seqh] likely remain at 4
- [qYVo] likely remain at 4
- [qYVo] likely remain at 2
- [qqrx] likely keep 6 or degrade to 4

---

### Decision · Program_Chairs · 2026-01-26

Reject